# Refined efficacy estimates of the Sanofi Pasteur dengue vaccine CYD-TDV using machine learning

I. Dorigatti [1], C.A. Donnelly[1], D.J. Laydon [1], R. Small[2], N. Jackson[3], L. Coudeville[4] & N.M. Ferguson [1]

CYD-TDV is the first licensed dengue vaccine for individuals 9–45 (or 60) years of age. Using 12% of the subjects enroled in phase-2b and phase-3 trials for which baseline serostatus was measured, the vaccine-induced protection against virologically confirmed dengue during active surveillance (0–25 months) was found to vary with prior exposure to dengue. Because age and dengue exposure are highly correlated in endemic settings, refined insight into how efficacy varies by serostatus and age is essential to understand the increased risk of hospitalisation observed among vaccinated individuals during the long-term follow-up and to develop safe and effective vaccination strategies. Here we apply machine learning to impute the baseline serostatus for subjects with post-dose 3 titres but missing baseline serostatus. We find evidence for age dependence in efficacy independent of serostatus and estimate that among 9–16 year olds, CYD-TDV is protective against serotypes 1, 3 and 4 regardless of baseline serostatus.

[1] MRC Centre for Global Infectious Disease Analysis, Department of Infectious Disease Epidemiology, School of Public Health, Imperial College London, Norfolk Place, London, W2 1PG, UK. [2] Sanofi Pasteur, 2501 Discovery Dr, Orlando, FL 32826, USA. [3] Sanofi Pasteur, 1541 Avenue Marcel Mérieux, 69280 Marcy l'Étoile, France. [4] Sanofi Pasteur, 14, Espace Henry Vallee, 690077 Lyon, France. Correspondence and requests for materials should be addressed to I.D. (email: i.dorigatti@imperial.ac.uk)

Dengue is a systemic viral infection caused by one of four closely related dengue viruses called serotypes (DENV1–4), and is the most prevalent arboviral infection in humans, with almost half of the world's population at risk of infection each year[1,2]. Due to the failure of traditional insecticide-based vector control strategies[3] and the absence of antiviral treatment, a dengue vaccine has been long pursued to curb dengue transmission and reduce the increasing disease and economic burden of dengue worldwide[4,5].

CYD-TDV, the dengue vaccine developed by Sanofi Pasteur and now marketed as Dengvaxia®, is the only dengue vaccine licensed to date[6]. Until now, CYD-TDV has been approved in more than 18 countries[7] and the first mass immunisation campaigns occurred in the Philippines and Brazil[8,9] in 2016 and 2017.

CYD-TDV is a live-attenuated recombinant tetravalent vaccine that uses the 17D yellow fever vaccine virus as a backbone and is administered in three doses given 6 months apart. Several trials have demonstrated the safe reactogenicity[10] and good immunogenicity profile of the vaccine against all serotypes[11,12]. A detailed analysis of multiple phase-2 trials of CYD-TDV revealed the fundamental role of dengue exposure prior to vaccination (herein referred to as baseline serostatus) on the vaccine immunogenicity and the ability of CYD-TDV to elicit a strong DENV4 antibody response since the first vaccine dose, that was comparable to the immunity observed upon natural infections in all subjects, including those with no evidence of dengue exposure before vaccination[12]. While the vaccine-induced antibody titres against DENV1–3 appeared lower than the respective antibody response elicited upon natural infection[12], the absence of correlates of protection implied that no conclusion on the expected vaccine efficacy could be drawn from the analysis of the immunogenicity data alone. In the phase-2b trial (CYD23, NCT00842530[13]), CYD-TDV was found to confer an imbalanced protection against the four serotypes, with non-significant efficacy observed against DENV2[14]. Two large-scale phase-3 trials conducted in Southeast Asia (CYD14, NCT01373281[15]) and Latin America (CYD15, NCT01374516[16]), later showed that beyond varying by serotype, efficacy against virologically confirmed disease depended on the baseline serostatus of the vaccine recipients, i.e. on the level of exposure to dengue at the time of vaccination according to the Plaque Reduction Neutralisation Test (PRNT50) results[17,18]. Based on 20% and 10% of the population enroled in CYD14 and CYD15, respectively (i.e. the trial population that by study design was serologically tested before the first vaccine dose was given), CYD-TDV was found to provide good protection against virologically confirmed dengue in subjects with prior exposure to dengue (74.3% [95% confidence interval (CI):53.2, 86.3%] in CYD14 and 83.7% [95% CI: 62.2, 93.7%] in CYD15) but lower and non-significant protection among baseline seronegative subjects (35.5% [95% CI: −26.8, 66.7%] in CYD14 and 43.2% [95% CI: −61.5, 80.0%] in CYD15)[17,18]. Pooled analysis of the results obtained in CYD14 and CYD15 showed that among children of 9–16 years of age, efficacy was 81.9% [95% CI: 67.2, 90.0%] in seropositive subjects and 52.5% [95% CI: 5.9, 76.1%] in seronegative subjects[19]. The increase in the relative risk of hospitalisation for virologically confirmed dengue in the vaccine group compared with the control group observed in the Southeast Asian trial (CYD14) among children 2–5 years old (relative risk 7.45[95% CI: 1.15–313.80]) and younger than 9 years (relative risk 1.58 [95% CI: 0.61–4.83]) during year 3 of the long-term follow-up phase of surveillance (24–36 months post first vaccination)[19], together with the less favourable efficacy results observed in young children[17,18], were key elements for choosing to license CYD-TDV among children of 9 years of age or older. New analyses of long-term data recently announced by Sanofi Pasteur[20,21] confirmed the differences in CYD-TDV performance according to baseline serostatus and resulted in a change in the vaccination recommendations both by Sanofi Pasteur and the World Health Organization (WHO)—with CYD-TDV now only being recommended for subjects exposed to dengue prior to the time of vaccination within the indicated age range[22]. The higher risk of severe dengue and dengue-related hospitalisation in baseline seronegative subjects up to 5 years post first vaccination reported by the WHO[23] indicate that there is a clear need for a better understanding of the role of age-specific effects, independent of serostatus. Due to a lack of power in the phase-2b and phase-3 trial data caused by the relatively small number of subjects tested at baseline, a full characterisation of how the efficacy of CYD-TDV varies by pre-exposure for further stratifications is lacking and there are uncertainties around existing efficacy estimates for baseline seronegative and baseline seropositive subjects[17,18].

Here we present the results of a post hoc analysis of the phase-2b (CYD23) and phase-3 (CYD14 and CYD15) trials of CYD-TDV to explore the extent to which machine learning and specifically a Boosted Regression Trees (BRT)[24] algorithm can be used to impute the missing information on the baseline serostatus for a subset of the subjects in the trials. By merging individual-level imputations with group-level inference of the baseline serostatus, we add precision to the estimates of how per-protocol efficacy (i.e. efficacy calculated on the subjects that received three vaccine doses) varies with serostatus, age and serotype during the active surveillance phase of the trials. We find that over all ages (2–16 years) and among 9–16 year olds, CYD-TDV is protective against serotypes 1, 3 and 4 regardless of baseline serostatus, while efficacy against serotype 2 is significant only for dengue pre-exposed subjects. Most notably, we find evidence for age dependence in efficacy independent of serostatus.

## Results

**Summary statistics of clinical trial data.** Table 1 provides a descriptive summary of the proportion of baseline seropositive subjects (among those with known baseline serostatus) observed in the phase-2b and phase-3 trials (CYD23, CYD14 and CYD15) during the active surveillance phase, stratified by age, arm, country of enrolment, gender, PD3 titre availability against any serotype, serotype of infection (for the cases) and trial. Table 2 summarises the results of the Pearson's chi-squared test that was conducted to assess the significance of the difference between the frequencies of cases and non-cases with known and unknown baseline serostatus across multiple stratifications. Over all ages, 3.6% (155/4251) of the subjects with known baseline serostatus and 4% (1242/29,527) of the subjects with unknown baseline serostatus were cases; the difference between these proportions is not statistically significant ($p$-value = 0.239, Pearson's chi-squared test). Among all other stratifications considered in Table 2, we found statistically significant differences in the number of cases and non-cases between subjects with known and unknown baseline serostatus in Honduras and in the Philippines ($p$-value = 0.001, Pearson's chi-squared test), in the Southeast Asian (CYD14) phase-3 trial ($p$-value = 0.019, Pearson's chi-squared test) and among subjects with known and unknown PD3 titres ($p$-values < 0.001, Pearson's chi-squared test).

**BRT optimisation and out-of-sample performance.** The out-of-sample sensitivity, specificity and percentage of correct predictions among cases, non-cases and overall obtained using 540 different model parameterisations from a grid search are

**Table 1 Summary of dengue exposure among cases and non-cases with known baseline serostatus**

| | Cases | | | Non-cases | | |
|---|---|---|---|---|---|---|
| | sero+ | sero− | %sero+ | sero+ | sero− | %sero+ |
| Age-group | | | | | | |
| All ages (2–16 years) | 90 | 65 | 58.1 | 3034 | 1062 | 71.4 |
| 2–8 years | 35 | 31 | 53 | 700 | 483 | 56.0 |
| 9–11 years | 28 | 20 | 58.3 | 1042 | 321 | 73.8 |
| 12–16 years | 27 | 14 | 65.9 | 1292 | 258 | 81.2 |
| Arm | | | | | | |
| CYD-TDV | 28 | 37 | 43.1 | 2079 | 695 | 73.2 |
| Placebo | 62 | 28 | 68.9 | 955 | 367 | 67.6 |
| Country | | | | | | |
| Brazil | 9 | 3 | 75.0 | 210 | 78 | 68.2 |
| Colombia | 15 | 8 | 65.2 | 829 | 61 | 90.8 |
| Honduras | 1 | 0 | 100.0 | 257 | 37 | 87.1 |
| Indonesia | 11 | 8 | 55.0 | 269 | 51 | 79.3 |
| Mexico | 4 | 3 | 57.1 | 167 | 147 | 52.0 |
| Malaysia | 2 | 4 | 33.3 | 139 | 152 | 46.8 |
| Philippines | 21 | 9 | 70.0 | 447 | 117 | 75.3 |
| Puerto Rico | 2 | 4 | 33.3 | 82 | 63 | 54.3 |
| Thailand (CYD23) | 7 | 6 | 53.8 | 203 | 84 | 67.7 |
| Thailand (CYD14) | 13 | 9 | 59.1 | 216 | 98 | 64.3 |
| Vietnam | 5 | 11 | 31.3 | 215 | 174 | 53.1 |
| Gender | | | | | | |
| Female | 40 | 29 | 58 | 1559 | 538 | 72.0 |
| Male | 50 | 36 | 58.1 | 1475 | 524 | 70.7 |
| PD3 titres against any serotype | | | | | | |
| Known | 79 | 48 | 62.2 | 2960 | 1032 | 71.9 |
| Unknown | 11 | 17 | 39.3 | 74 | 30 | 56.1 |
| Serotype | | | | | | |
| DENV1 | 27 | 26 | 50.9 | – | – | – |
| DENV2 | 20 | 17 | 54.1 | – | – | – |
| DENV3 | 21 | 9 | 70 | – | – | – |
| DENV4 | 18 | 11 | 62.1 | – | – | – |
| Unknown | 4 | 2 | 66.7 | – | – | – |
| Trial | | | | | | |
| CYD23 (phase-2b) | 7 | 6 | 53.8 | 203 | 84 | 67.7 |
| CYD14 (phase-3) | 52 | 41 | 55.9 | 1286 | 592 | 65.2 |
| CYD15 (phase-3) | 31 | 18 | 63.3 | 1545 | 386 | 78.0 |

Cases and non-cases occurred during the active surveillance phase (up to 25 months after the first dose) in the phase-2b and phase-3 trials (CYD23, CYD14 and CYD15), stratified by baseline serostatus, age, arm, country of enrolment, gender, PD3 titre availability, serotype of infection (for cases) and trial. sero+ stands for baseline seropositive; sero− stands for baseline seronegative. In each stratification the percentage of baseline seropositive subjects (%sero+) was calculated by dividing the number of seropositive subjects by the total number of seropositive and seronegative subjects

with BRT are robust to the choice of the searching algorithm adopted (i.e. random versus grid search) and the specific optimal parameters used (Supplementary Figure 10).

Automatic feature selection indicated that trial and case occurrence could be dropped from the predictors with minimal (<0.17%) changes in the out-of-sample prediction accuracy (Supplementary Table 2 and Supplementary Figure 11). Sensitivity analysis on the accuracy of a more parsimonious model with serotype, time interval between dose 3 and symptoms onset and gender removed on top of trial and case occurrence (Supplementary Figure 11) showed a small yet significant (1.3%) drop in performance especially among cases (from 74.2% to 72.9%), which suggested that retention of serotype, time interval between dose 3 and symptoms onset and gender as predictors in the final model was adequate.

**Vaccine efficacy estimates by baseline serostatus.** Supplementary Figure 12 shows that the imputed baseline seroprevalence (proportion of individuals seropositive to dengue among those with observed or imputed baseline serostatus) is consistent with the observed baseline seroprevalence for both cases and non-cases. The corresponding vaccine efficacy estimates are shown in Fig. 1 and Supplementary Figure 13. Among baseline seropositive subjects, we generally estimate high efficacy that is significantly positive for the overall population (Fig. 1b) and for population stratifications by age (Fig. 1c) or by infecting serotype (Fig. 1d–f). The only exceptions to this trend are the efficacy against DENV1 and DENV2 in children 2–8 years old (Fig. 1e). The estimated vaccine efficacy among baseline seropositive subjects was significantly positive in all countries (Supplementary Figure 13b).

Among baseline seronegative subjects, we estimate lower but still positive efficacy overall (Fig. 1b). However, this result is not as robust across subdivisions in the data. For seronegative subjects, efficacy is estimated to be significantly positive among children ≥9 years (Fig. 1c), against DENV1, DENV3 and DENV4 over all ages (Fig. 1d) and against DENV1, DENV3 and DENV4 among 9–16-year olds (Fig. 1f). We find that vaccine efficacy is not significantly positive in 2–8-year-old children who are seronegative at baseline, both over all serotypes pooled (Fig. 1c) and against DENV1, DENV2 and DENV4 (Fig. 1e). The estimated vaccine efficacy was non-significantly negative against DENV2 using all stratifications tested (Fig. 1d–f and Supplementary Figure 13d–e). Among all study subjects (2–16 years), we find significantly different vaccine efficacies against DENV2 and DENV4 between baseline seropositive and baseline seropositive subjects (Table 3). However, these differences were not statistically significant when calculated among children 2–8 and 9–16 years old separately (Table 3). The low numbers of DENV2 cases observed among 9–16-year olds and DENV2 and DENV4 cases observed among 2–8 years old drive the wider uncertainties observed in the relative efficacies. Large variations are also seen between countries in estimated vaccine efficacy among baseline seronegative subjects (Supplementary Figure 13b), potentially due to the relatively small number of baseline seronegative subjects compared to seropositive subjects enroled in the trials, reflecting the highly endemic transmission of dengue in the trial locations. Within Latin America, all countries except Mexico and Puerto Rico show significantly positive vaccine efficacy among baseline seronegative subjects, varying from a high of 54% [95% CI: 22, 88%] in Honduras to a low of −9% [95% CI: −87, 35%] in Mexico (Supplementary Figure 13b). Within Southeast Asia, all countries except Indonesia, Thailand and Vietnam show significantly positive vaccine efficacy among baseline seronegative subjects, with an average efficacy varying

presented in Supplementary Figures 1–9. BRT models trained on >2000 subjects performed best, regardless of the tuning parameters (Supplementary Figures 1–9). Overall, the accuracy was maximised for intermediate values of tree complexity (tc), tending to increase with smaller learning rates (lr), and was higher when sampling variability was included (bf = 0.5, 0.75). The highest accuracy among cases and non-cases separately was achieved for BRT models trained on 50% of non-cases and 75% of cases, using tc = 16, lr = 0.0005 and bf = 0.75; this parameterisation was adopted for the rest of the analysis. Sensitivity analysis on the performance of a random search to identify the optimal parameterisation showed that the accuracies achieved

**Table 2 Summary proportions of cases with known or unknown baseline serostatus across stratifications**

| | Known baseline serostatus | | | Unknown baseline serostatus | | | *p*-value |
|---|---|---|---|---|---|---|---|
| | Cases | Non-cases | %cases | Cases | Non-cases | %cases | |
| **Age-group** | | | | | | | |
| All ages (2–16 years) | 155 | 4096 | 3.6 | 1242 | 29,527 | 4.0 | 0.239 |
| 2–8 years | 66 | 1183 | 5.3 | 405 | 6073 | 6.3 | 0.213 |
| 9–11 years | 48 | 1363 | 3.4 | 473 | 11,703 | 3.9 | 0.412 |
| 12–16 years | 41 | 1550 | 2.6 | 364 | 11,751 | 3.0 | 0.385 |
| **Arm** | | | | | | | |
| CYD-TDV | 65 | 2774 | 2.3 | 579 | 19,943 | 2.8 | 0.118 |
| Placebo | 90 | 1322 | 6.4 | 663 | 9584 | 6.5 | 0.936 |
| **Country** | | | | | | | |
| Brazil | 12 | 288 | 4.0 | 107 | 3137 | 3.3 | 0.633 |
| Colombia | 23 | 890 | 2.5 | 250 | 8555 | 2.8 | 0.651 |
| Honduras | 1 | 294 | 0.3 | 112 | 2379 | 4.5 | **0.001** |
| Indonesia | 19 | 320 | 5.6 | 64 | 1447 | 4.2 | 0.339 |
| Mexico | 7 | 314 | 2.2 | 128 | 3000 | 4.1 | 0.126 |
| Malaysia | 6 | 291 | 2.0 | 24 | 1075 | 2.2 | 1 |
| Philippines | 30 | 564 | 5.1 | 265 | 2611 | 9.2 | **0.001** |
| Puerto Rico | 6 | 145 | 4.0 | 18 | 1145 | 1.5 | 0.076 |
| Thailand (CYD23) | 13 | 287 | 4.3 | 122 | 3580 | 3.3 | 0.429 |
| Thailand (CYD14) | 22 | 314 | 6.5 | 68 | 759 | 8.2 | 0.396 |
| Vietnam | 16 | 389 | 4.0 | 84 | 1839 | 4.4 | 0.809 |
| **Gender** | | | | | | | |
| Female | 69 | 2097 | 3.2 | 627 | 15,078 | 4.0 | 0.078 |
| Male | 86 | 1999 | 4.1 | 615 | 14,449 | 4.1 | 0.974 |
| **PD3 titres against any serotype** | | | | | | | |
| Known | 127 | 3992 | 3.1 | 658 | 19 | 97.2 | **<0.001** |
| Unknown | 28 | 104 | 21.2 | 584 | 29,508 | 1.9 | **<0.001** |
| **Serotype** | | | | | | | |
| DENV1 | 53 | – | – | 410 | – | – | – |
| DENV2 | 37 | – | – | 370 | – | – | – |
| DENV3 | 30 | – | – | 217 | – | – | – |
| DENV4 | 29 | – | – | 201 | – | – | – |
| Unknown | 6 | – | – | 44 | – | – | – |
| **Trial** | | | | | | | |
| CYD23 (phase-2b) | 13 | 287 | 4.3 | 122 | 3580 | 3.3 | 0.429 |
| CYD14 (phase-3) | 93 | 1878 | 4.7 | 505 | 7731 | 6.1 | **0.019** |
| CYD15 (phase-3) | 49 | 1931 | 2.5 | 615 | 18,216 | 3.3 | 0.066 |

Cases and non-cases occurred during the active surveillance phase (up to 25 months after the first dose) in the phase-2b and phase-3 trials (CYD23, CYD14 and CYD15), stratified by baseline serostatus, age, arm, country of enrolment, gender, PD3 titre availability, serotype of infection (for cases) and trial. In each stratification the percentage of cases were calculated by dividing the number of cases by the total number of cases and non-cases. The *p*-values were obtained with the Pearson's chi-squared test using the number of cases and non-cases with known and unknown baseline serostatus in each stratification. Significant (<0.05) *p*-values are in bold

from a high of 67% [95% CI: 32, 91%] in Malaysia to a low of 7% [95% CI: −74, 54%] in Indonesia (Supplementary Figure 13b). Among baseline seronegative subjects, in Southeast Asia efficacy against DENV2, DENV3 and DENV4, and efficacy among 2–5-year-old children are non-significant (Supplementary Figures 13d–f); in Latin America we find non-significant efficacy only against DENV2 and DENV4 (Supplementary Fig. 13e).

In Supplementary Table 12 we present the percentage increases in variance in vaccine efficacy due to missing baseline serostatus among the subjects with observed PD3 titres.

Figure 2 shows the vaccine efficacy estimates for the finer stratification of baseline serostatus into seronegative, monotypic and multitypic categories (see Methods for the definitions used). Additional results and sensitivity analysis on alternative definitions of monotypic and multitypic PRNT50 profiles are given in Supplementary Figures 22–39.

**Sensitivity analysis.** Sensitivity analysis on the impact of group-level inference on the efficacy estimates shows that individual-level imputation of the baseline serostatus for 677 subjects alone significantly reduces the uncertainty around the efficacies calculated on the subjects with observed baseline serostatus (Supplementary Figures 18 and 19). Inference of baseline serostatus at the group-level appears to slightly increase average efficacy estimates for DENV2 among baseline seronegative subjects, although this does not affect the overall interpretation and consistency of the results.

Discarding the phase-2b data from the vaccine efficacy estimation but not BRT model training (Supplementary Figure 20) or removing the phase-2b data entirely (Supplementary Figure 21) does not substantially affect the efficacy estimates; the only notable difference is that the negative lower bound of efficacy against DENV3 among seronegative 2–8-year olds (Supplementary Figures 20e and 21e) is positive when the phase-2b data are included (Fig. 1e).

**Age-trend in vaccine efficacy.** When imputed data on baseline serostatus were not used in estimating efficacy, we found a significant age-trend in the vaccine efficacy among baseline seronegative subjects using 2–8, 9–11 and 12–16 years age-groups ($p$-value = 0.04, $F$-test), giving an increase in vaccine efficacy of 4.6% [95% CI: 0.4, 8.9%] for each year increase in

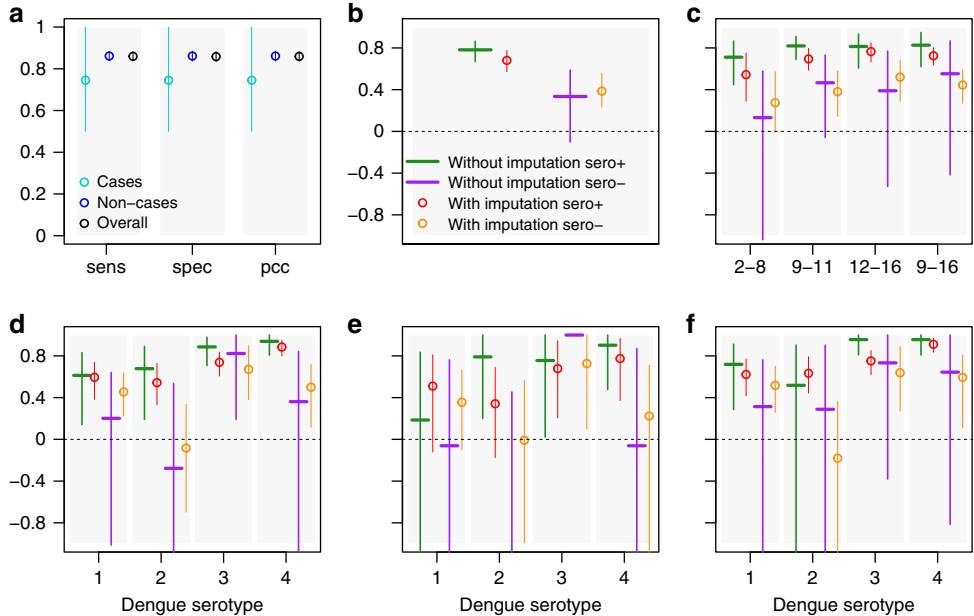

**Fig. 1** Mean and 95% CI of vaccine efficacy estimates with and without imputations for baseline seronegative (sero−) and baseline seropositive (sero+) subjects. Estimates were obtained from 1000 realisations of the final BRT model trained on 50% of non-cases and 75% of cases and using 10-fold cross-validation, tc = 16, lr = 0.0005, bf = 0.75. **a** Sensitivity (sens), specificity (spec) and proportion of correct classifications (pcc) among cases, non-cases and overall. **b** Vaccine efficacy estimates for baseline seropositive and baseline seronegative subjects separately. **c** Vaccine efficacy estimates for baseline seropositive and baseline seronegative subjects by age using 2–8, 9–11, 12–16 and 9–16 years age groups. **d** Vaccine efficacy estimates by serotype for baseline seropositive and baseline seronegative subjects of all ages (2–16 years). **e** Vaccine efficacy estimates by serotype for baseline seropositive and baseline seronegative subjects 2–8 years old. **f** Vaccine efficacy estimates by serotype for baseline seropositive and baseline seronegative subjects 9–16 years old

age. This trend became non-significant (*p*-value > 0.05, *F*-test) when estimates included imputed baseline serostatus data. However, using estimates that included imputed baseline serostatus data and 2–5, 6–11 and 12–14 age-groups, we found a consistent (4.7% [95% CI: 0.05, 9.3%], *p*-value = 0.05, *F*-test) increase in vaccine efficacy for each year increase in age among baseline seronegative children in Southeast Asia. The significance and consistency of the age-trend in vaccine efficacy among baseline seronegative subjects was further confirmed by the results of weighted linear regression on the vaccine efficacy estimates obtained with imputation using all trials and a finer age-stratification into 2-year age-groups (i.e. 2–3, 4–5, 6–7, 8–9, 10–11, 12–13 and 14–16 years), where we found 2.9% [95% CI: 0.4, 5.4%] (*p*-value = 0.03, *F*-test) increase in vaccine efficacy for each year-increase in age. Additional results on the age-trend in vaccine efficacy obtained with and without imputation are given in Supplementary Tables 7–11 and in Supplementary Figures 15–17.

**Association between age and serotype of infection.** The lower efficacy estimates seen for 2–8-year olds and against DENV2 prompted us to test the significance of the association between age and the serotype of infection. Without stratifying by baseline serostatus and using either 2–8, 9–11, 12–16 or 2–8, 9–16 age-groups, we found a significant association (*p*-values < 0.0001, Fisher's exact test), with more DENV2 and fewer DENV3 cases in the 2–8 age-group than expected and more DENV3 cases among 9–11 and 9–16-year olds than expected if age and serotype were independent (Supplementary Tables 3–6). These associations were non-significant examining seronegative and seropositive subjects separately if imputed data on baseline serostatus were not used. However, including imputed data

(i.e. by running 1000 realisations of the final BRT model), we found that around 80% of realisations yielded a significant association (*p*-value < 0.05, Pearson's chi-squared test) between age and the serotype of infection for both seropositive and seronegative subjects separately, with more DENV2 and fewer DENV3 cases than expected among seropositive subjects in the 2–8 years age-group and more DENV3 cases than expected among seronegative subjects in the 9–11 years age-group (Supplementary Figure 14).

**Discussion**

The statistically significant differences in the proportions of cases and non-cases between subjects with known and unknown baseline serostatus in Honduras, the Philippines (*p*-value = 0.001, see Table 2) and in the Southeast Asian (CYD14) phase-3 trial (*p*-value = 0.019, see Table 2) are likely due to the lack of full randomisation in the assignment of a pre-defined number of subjects to the immunogenicity subsets (i.e. the subset of subjects with known baseline serostatus) in each site of the trials[15,16], which were instead established according to the time of subject enrolment in the trials, in a chronological fashion. The statistically significant difference observed in the proportions of cases with known or unknown baseline serostatus, among both subsets of subjects with known and unknown PD3 titres (*p*-values < 0.001, see Table 2), was due to the fact that the trial design specified that PD3 blood samples were retrospectively tested for dengue antibody levels for all dengue cases plus all participants in the immunological subsets of the trials. Our analysis shows that BRT, a machine learning algorithm, can impute the baseline serostatus of subjects with observed PD3 PRNT50 titres with high accuracy, using a dichotomous classification of the subjects into

**Table 3 Vaccine efficacy estimates by baseline serostatus, serotype and age-group obtained with imputation**

| | All ages (2–16 years) | | | 2–8-year olds | | | 9–16 year olds | | |
|---|---|---|---|---|---|---|---|---|---|
| | sero+ | sero− | *p*-value | sero+ | sero− | *p*-value | sero+ | sero− | *p*-value |
| DENV1 | 0.59 (0.38, 0.74) | 0.45 (0.22, 0.64) | 0.309 | 0.51 (−0.12, 0.81) | 0.36 (−0.10, 0.66) | 0.624 | 0.62 (0.42, 0.77) | 0.52 (0.26, 0.70) | 0.469 |
| DENV2 | **0.54 (0.33, 0.73)** | **−0.08 (−0.70. 0.38)** | **0.029** | 0.34 (−0.17, 0.69) | −0.01 (−0.99, 0.56) | 0.426 | 0.63 (0.44, 0.79) | −0.18 (−1.41, 0.36) | 0.085 |
| DENV3 | 0.74 (0.61, 0.84) | 0.67 (0.38, 0.90) | 0.630 | 0.68 (0.21, 0.95) | 0.73 (0.10, 1.00) | 0.885 | 0.75 (0.62, 0.85) | 0.64 (0.27, 0.89) | 0.485 |
| DENV4 | **0.88 (0.80, 0.95)** | **0.50 (0.12, 0.72)** | **0.019** | 0.77 (0.37, 0.97) | 0.22 (−1.28, 0.356) | 0.379 | 0.91 (0.84, 0.97) | 0.59 (0.11, 0.81) | 0.070 |

Mean and 95% CI (within parentheses) of the vaccine efficacy estimates by baseline serostatus, serotype and age-group generated with imputation and significance of the statistical difference in vaccine efficacy between baseline seropositive and baseline seronegative subjects against each serotype in each age-group. The *p*-values were obtained using the Pearson's chi-squared test. sero+ stands for seropositive; sero− stands for seronegative. Significant (<0.05) *p*-values and the corresponding estimates are in bold

seronegative/seropositive profiles among cases (mean 75% [95% CI: 44, 100%]) and among non-cases (mean 86% [95% CI: 84, 88%]). Analysis of the accuracy achieved by BRT compared to other commonly adopted machine learning algorithms (including generalised linear models, random forest and neural networks) shows the optimal predictive performance of BRT on the data analysed in this study (Supplementary Figures 42 and 43 and Supplementary Table 14). Although conducted on a minority (2.2%) of the subjects with missing baseline serostatus (i.e. the 677 subjects with observed PD3 PRNT50 titres), data imputation greatly reduced the uncertainty around the efficacy estimates, both on its own (Supplementary Figures 18 and 19) and when coupled with group-level inference on the baseline exposure (Fig. 1 and Supplementary Figure 13). The increased precision of the vaccine efficacy estimates obtained with imputation suggests significantly positive efficacy among baseline seronegative children (i) 9–16 years when pooling serotypes, (ii) 9–16 years against DENV1, DENV3 and DENV4, (iii) against DENV1, DENV3 and DENV4 over all ages (Fig. 1 and Supplementary Figure 18), (iv) in both CYD14 and CYD15 and (v) in Brazil, Colombia, Malaysia and the Philippines (Supplementary Figures 13 and 19), where the efficacy estimates in the absence of imputation were not significantly positive. These results are reassuring in the context of the past immunisation campaign conducted in the Philippines[9] in 2017, which were stopped following the press release announcing the results obtained from new analysis of long-term data[20]. However, our estimates obtained with imputation suggest non-significant efficacy against DENV2 among seronegative subjects using a variety of age-stratifications (2–16 (all ages), 2–8 and 9–16 years) and among seropositive subjects in the 2–8 years age-group. Several potential factors may explain this finding, including the suggested higher propensity of DENV2 to cause symptomatic/severe disease[25,26] possibly linked with its association with secondary infection[27,28], higher levels of neutralising antibodies needed to confer protection against DENV2[29] and the lack of dengue non-structural proteins in the vaccine formulation, which may be particularly relevant given the specific targeting of non-structural proteins observed in the T-cell responses following natural DENV2 infection[30].

We found significant age dependence in vaccine efficacy estimates, independent of the baseline serostatus (both with and without imputation of the baseline serostatus), indicating that older children benefit more from vaccination with CYD-TDV than younger children. Our estimates imply that a 10-year age difference in the age of vaccination may confer on average 46% [95% CI: 4, 89%] higher protection against virologically confirmed dengue among baseline seronegative subjects. Age dependence in efficacy could be due to maturation of the immune system[31,32], age dependence in dengue infection severity[33] or a potential role of Japanese encephalitis (JE) or yellow fever (YF)

virus exposure, either due to natural infection or vaccination. This latter hypothesis is consistent with the analysis conducted by Dorigatti et al.[12], where pre-existing immunity to JE was shown to induce a broader and stronger response to CYD-TDV vaccination. Similarly, vaccination with CYD-TDV in YF pre-immune individuals could recall cellular responses against the YF backbone strain, which could assist through a bystander effect on specific responses against the envelope proteins. Unfortunately, these assumptions could not be further validated due to the unavailability of information on pre-existing JE/YF vaccinations and on the JE/YF baseline serostatus of the subjects in this study. This information could also have potentially improved the predictive performance of the statistical models developed in this study.

In future work it will be interesting to apply the method developed in this paper to estimate the efficacy of CYD-TDV using the long-term follow-up data of the phase-2b and phase-3 trials. This would allow us to explore whether efficacy varies in time, whether any temporal changes in efficacy depend on the baseline serostatus and the relative contribution of temporal clustering[34] versus vaccine-induced immunological priming[35,36] to the higher relative risks of hospitalisation seen among 2–5-year olds in the first two years of passive surveillance[33]. In addition, it will be important to assess how to optimise vaccine deployment[35,37] given the heterogeneity in efficacy between serotypes and ages and how this varies with baseline serostatus.

The spatially and temporally heterogeneous dynamics of dengue circulation and serotype replacement, which is often exhibited at the macro as well as at the micro scale[38–41], may explain the observed association between age and the serotype of infection (more DENV2 cases and fewer DENV3 cases than expected observed among 2–8-year olds). However, the stronger association between DENV2 infections and pre-exposure to dengue is consistent with previous observations[42,43].

Using a more refined partitioning of baseline seropositive subjects into monotypic (a single PRNT50 titre > 10 or, if more than one PRNT50 titre > 10 are present, a single PRNT50 titre > 80) or multitypic PRNT50 profiles, we found that BRT models achieved lower accuracy both among cases (mean of percentage of correct classifications 71% [95% CI: 43–100%]) and non-cases (74% [95% CI: 70–78%]), with lower performance observed among baseline monotypic profiles, especially non-cases. This lower performance can be attributed to (i) the low prevalence of monotypic profiles in the data, i.e. 817 out of the 4251 (19.2%) subjects with observed baseline serostatus using the main definition (334 out of the 4251 (7.8%) subjects when using the alternative definition of monotypic subjects as those with only one PRNT50 titre > 10, Supplementary Figures 35–39), (ii) the similarities of the baseline monotypic PD3 PRNT50 titres with baseline seronegative profiles (Supplementary Figures 40 and 41), which is particularly relevant given the high (74.6%) relative importance of the PD3 titres in determining the

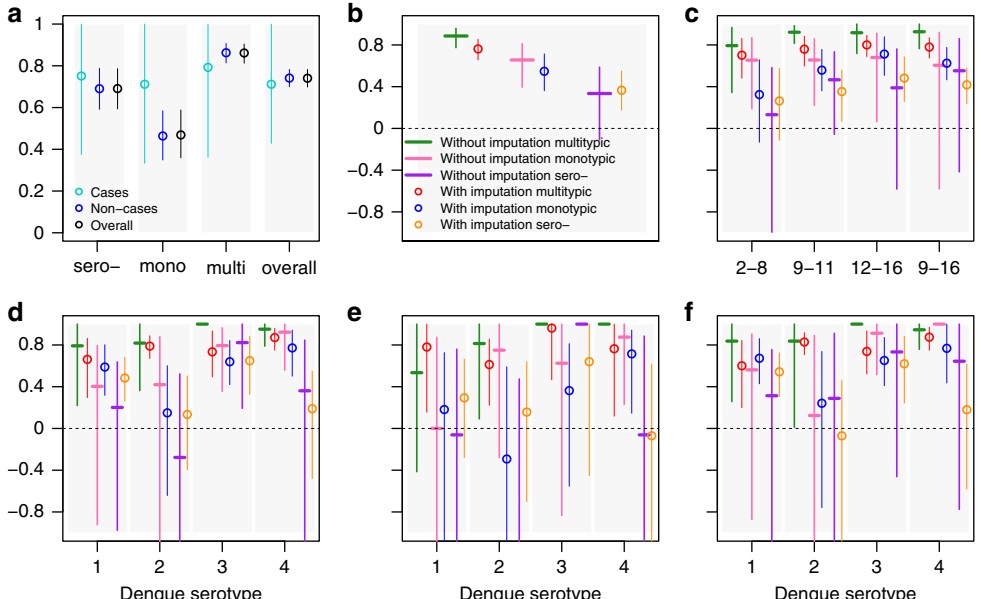

**Fig. 2** Mean and 95% CI of vaccine efficacy estimates with and without imputations for subjects with baseline seronegative, monotypic and multitypic PRNT50 titre profiles. Estimates were obtained from 1000 realisations of the final BRT model achieving at least 30% accuracy for cases and non-cases, trained on 75% of non-cases and 75% of cases and using five-fold cross-validation, $tc = 15$, $lr = 0.001$, $bf = 0.75$. **a** Proportion of correct classifications among seronegative (sero−), monotypic (mono), multitypic (multi) and overall (i.e. seronegative, monotypic and multitypic) cases, non-cases and overall (cases and non-cases). **b** Vaccine efficacy estimates for baseline multitypic, monotypic and seronegative subjects separately. **c** Vaccine efficacy estimates for baseline multitypic, monotypic and seronegative subjects by age, using 2–8, 9–11, 12–16 and 9–16 years age-groups. **d** Vaccine efficacy estimates by serotype for baseline multitypic, monotypic and seronegative subjects over all ages (2–16 years). **e** Vaccine efficacy estimates by serotype for baseline multitypic, monotypic and seronegative subjects 2–8 years old. **f** Vaccine efficacy by serotype for baseline multitypic, monotypic and seronegative subjects 9–16 years old

baseline serostatus (Supplementary Table 13) and (iii) the choice of the hyper-parameterisation so as to maximise the predictive accuracy among cases, driven by the large prevalence of cases (98.6%, i.e. 658 cases out of 677 subjects with PD3 titres) in the prediction set.

In general, we found that the vaccine efficacy among baseline monotypic subjects lies between the efficacy estimated for baseline multitypic and baseline seronegative subjects, which is in good agreement with the immunogenicity patterns observed in the phase-3, phase-2b and multiple phase-2 trials[12]. Interestingly, we find significantly positive vaccine efficacy among baseline monotypic profiles for children over 9 years over all serotypes when pooled together, and against DENV1, DENV3 and DENV4 but not against DENV2 individually. Using a finer stratification of the baseline serostatus into seronegative, monotypic and multitypic profiles reduces the precision of both the algorithm's accuracy and efficacy estimates, especially among the subjects with a monotypic profile, which consistently account for the minority of the trials subjects. The wide uncertainty around the efficacy estimates obtained for the baseline monotypic profiles limits the extent to which these estimates can be used to inform future vaccination strategies. However, the results of the ongoing long-term follow-up phase of the phase-3 and phase-2b clinical trials together with the results of post-marketing surveillance will provide crucial information on the serotype-specific safety, efficacy and effectiveness of CYD-TDV not only among baseline seronegative subjects but also among baseline monotypic profiles.

The use of machine learning and in particular BRT for data imputation in the context of vaccine trials is novel, although imputation using BRT has been widely applied in other contexts, such as species distribution in ecology[24,44,45] and

spatial epidemiology[2,46,47]. Typically, multiple imputation in vaccine trials is conducted by building a single predictive model which is used to impute the missing data several times and then using Rubin's rule to integrate the results. Here we bootstrapped (i.e. sampled with replacement) the whole dataset several times, calculated a BRT model at each time and then combined the results. Further analyses of the data used in this study, which employ different imputation methods, are ongoing and it will be interesting to test whether the heterogeneities and trends obtained in this study are consistent across imputation methods.

While alternative therapeutics (e.g. antivirals and monoclonal antibodies) are in the pipeline[48] and other dengue vaccine technologies[49] are in development, the Sanofi-Pasteur vaccine remains the only tool currently available to target the disease and economic burden of dengue in high-transmission settings. In this paper, we have provided the most refined characterisation of CYD-TDV's efficacy profile available to date and have shown that machine learning is a useful tool to tackle the statistical challenges associated with missing or incomplete data in the analysis of clinical trials.

## Methods
**Data**. We analysed the results of the phase-2b (CYD23) and phase-3 (CYD14 and CYD15) clinical trials of CYD-TDV during active surveillance (up to 25 months after the first dose), including 35,020 subjects overall. For all subjects, information on the trial (CYD23, CYD14 or CYD15), country of enrolment (Indonesia, Malaysia, Philippines, Thailand, Vietnam, Brazil, Colombia, Honduras, Mexico, Puerto Rico), arm (vaccine or placebo), age (between 2 and 16 years, continuous variable), gender (female/male), presence of virologically confirmed dengue disease and, for the cases, infecting serotype (1, 2, 3, 4 or untyped) and time interval between dose 3 and symptoms onset (in days, continuous variable) was available.

Over all trials, 4251 out of 35,020 subjects (12%) had baseline (i.e. before first vaccination) antibody titres against DENV1–4 quantified by Plaque Reduction Neutralisation Test (PRNT50). Among the 4251 subjects with such baseline serostatus data, 4119 (96.8%) had observed post-dose 3 (PD3) PRNT50 titres against DENV1–4. This latter number excludes vaccine recipients with symptomatic dengue manifestations occurring before dose 3, for which the PD3 PRNT50 titres were considered unknown. Data imputation was performed for 677 subjects who had missing baseline status and observed PD3 PRNT50 titres. Tables 1 and 2, respectively, provide descriptive summaries of the number of subjects seropositive/seronegative at baseline and of the number of cases and non-cases observed in the phase-2b and phase-3 trials (CYD23, CYD14 and CYD15) during the active surveillance phase, stratified by age, arm, country of enrolment, gender, PD3 titre availability against any serotype, serotype of infection (for the cases) and trial. Additional statistics on the dataset are provided in Supplementary Methods (see Section 1.1) and further descriptive summaries of the total number of cases infected by serotype, age-group and baseline serostatus are provided in Supplementary Tables 3–6.

**BRT optimisation and vaccine efficacy estimation**. In our primary analysis, individuals with baseline PRNT50 titres < 10 (1/dil) were classified as baseline seronegative, whereas individuals with at least one baseline PRNT50 titre ≥ 10 were considered baseline seropositive. We trained BRT models to predict the baseline dengue serostatus using the PD3 titres, trial, country, age, gender, case occurrence during active surveillance, the infecting serotype (for cases) and the time interval between dose 3 and symptoms onset (for cases) as predictors. BRT models were trained on the 4,251 subjects with observed baseline serostatus, where subjects were randomly assigned to the training or out-of-sample validation set. We tested 9 different sizes of the training set and 60 combinations of the tuning parameters of the BRT model for a total of 540 scenarios (Supplementary Methods, Section 1.2.1). Each scenario was tested on 100 training and validation sets randomly sampled without replacement. BRT models were built using 10-fold cross-validation using the deviance as the loss function (additional information is provided in Supplementary Methods, Section 1.2.2). The optimal size of the training set and the optimal set of tuning parameters were determined with reference to the out-of-sample sensitivity (fraction of baseline seropositive subjects correctly classified), specificity (fraction of baseline seronegative subjects correctly classified) and percentage of predictions correctly classified overall and among cases and non-cases occurring during the active phase of surveillance separately (Supplementary Figures 1–9). We then performed variable elimination using automatic feature selection to drop the variables that gave no evidence of improving the predictive performance of the model[24,50,51] (Supplementary Table 2) and obtained a final model which was used to impute the baseline dengue serostatus of the 677 subjects with missing baseline serostatus but observed PD3 PRNT50 titres. Subjects with observed and imputed baseline dengue serostatus were used to infer the group-level baseline serostatus of 30,282 subjects who had missing PD3 titres and missing baseline serostatus (details are provided in Supplementary Methods, Section 1.2.4). We then estimated the vaccine efficacy by baseline exposure status using all subjects enroled in the trials.

Efficacy estimates by baseline serostatus obtained with and without imputed data were calculated across subjects who received three vaccine doses (per-protocol population). Confidence intervals were generated by bootstrapping. For consistency, sampling variability was included in the BRT models to produce vaccine efficacy estimates (for details see Supplementary Methods, Section 1.2.5).

Age-trends in vaccine efficacies by pre-exposure were tested using weighted linear regression, using the average efficacies for each age-group as the response variable, the mid-point of the age-groups as the predictor and the reciprocal of the efficacies' variance as the weight. We used the Pearson's chi-squared test with significance level 0.05 to test for trends in vaccine efficacy between two age-groups, and the Fisher's exact test to test the association between the age-group of cases and the serotype of infection.

In a secondary analysis we tested the performance of BRT using a finer classification of seropositive subjects into monotypic or multitypic PRNT50 profiles. Baseline seropositive individuals with a single PRNT50 titre > 10 or, if more than one PRNT50 titre > 10 were present, a single PRNT50 titre > 80 were classified as monotypic, otherwise as multitypic. Results of a sensitivity analysis conducted on the definitions used are presented in Supplementary Figures 35–39.

**Ethical compliance statement**. The authors confirm that the clinical trials analysed in this paper (registration numbers: NCT00842530[13] (CYD23), NCT01373281[15] (CYD14) and NCT01374516[16] (CYD15)) comply with all relevant ethical regulations and have obtained informed consent by all study participants.

**Code availability**. The analyses presented in this study were conducted in the statistical software R version 3.3.2[52] using the dismo[53], gbm[54] and PresenceAbsence[55] packages. The computer code used to generate the results reported in this study is available from the authors upon request.

**Data availability**
The data that support the findings presented in this paper were obtained under license for the current study and are not publicly available. Data are however available from the authors upon reasonable request and with permission of Sanofi Pasteur. Details on Sanofi Pasteur's data sharing criteria, eligible studies, and on the process for requesting access to anonymized patient level data and related study documents including clinical study report, study protocol with any amendments, blank case report form, statistical analysis plan, and dataset specifications can be found at https://www.clinicalstudydatarequest.com.

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

## Acknowledgements

The authors express their gratitude to the investigators and subjects that participated to the CYD-TDV clinical trials and thank Dr. Samir Bhatt for useful discussion. I.D., C.A.D., D.J.L. and N.M.F. acknowledge research funding from the Imperial College Junior Research Fellowship scheme, the Bill and Melinda Gates Foundation, the National Institute of General Medical Sciences (NIGMS) 'Models of Infectious Disease Agent Study' (MIDAS) initiative and the UK Medical Research Council.

## Author contributions

I.D. and N.M.F. conceived the study; I.D., C.A.D., R.S. and N.M.F. conceived the statistical model; I.D. analysed the data; I.D. wrote the manuscript; all authors contributed to the interpretation of the result and reviewed the manuscript.

## Additional information

**Competing interests:** R.S., N.J. and L.C. are employed by Sanofi Pasteur, the producer of CYD-TDV. The remaining authors declare no competing interests.

