## [Peer Review File · Nature Communications]

Reviewers' comments:

Reviewer #1 (Remarks to the Author):

This research seeks to provide "a full chracterization of how the efficacy of CYD-TDV varies by pre-exposure". Previous studies have not been able to do this due to the fact that only 12% of subjects in all trials had baseline serostatus data. In this paper optimised boosted regression trees (BRT) are used to impute the missing baseline serostatus data, allowing a more powerful analysis than was possible previously. Great care was taken to optimise the final BRT model and the relative importances for the predictor variables are useful.

However, further details are required in order to provide more confidence in the results relating to the imputation and the vaccine efficacy estimates. Ideally this evidence should be provided in the paper itself rather in what is a very long and probably not ideal set of supplementary materials. Figure S1-S9 are particularly difficult to read.

It is suggested that the authors provide detailed descriptive data regarding the number of (dengue) cases and non-cases for each age group. This should be done separately for the data for which baseline serostatus was known and where it was unknown. A comparison of these two data sets is also required in terms of other variables such as gender, age, country etc.

It is being assumed that one can impute baseline serostatus for 88% of the subjects from only 12% of the subjects, so some assurance is required as to the similarity of these two data sets in order to justify the whole process of baseline serostatus imputation. The relatively small number of (dengue) cases (155) among the 4251 subjects with baseline serostatus makes this descriptive analysis especially important. What is the percetnage of dengue cases for the remaining data? What percentage of these subjects are sero+ and sero- for each of the data sets, broken down by age group. Even powerful bootstrap methods have limited value when there are very small numbers of (dengue) cases.

The methods section reports weighted linear regression analyses and Pearson Chi-Square tests for two or three age groups. I could not find details about these analyses in the supplementary materials. Using average efficacies rather than the subject data itself seems to be insufficient. On the other hand Pearson Chi-Squared tests, if used with subject data, would be unrelaible with such large sample sizes. These analyses are critical in terms of the conclusions, so more convincing analyses and explanations are required for the vaccine efficacy results.

No detailed hypothesis testing results have been supplied. Instead there are 12 graphs with a somewhat confusing layout.

A lot of work has gone into this paper. It is hoped that the above feedback will make it possible for future readers to better appreciate its value.

Reviewer #2 (Remarks to the Author):

This reviewer gives a reject to this paper.

This paper describes the approach of building a machine learning model to fill in missing values to better estimate the efficacy of CYD-TDV, a dengue vaccine. This is an interesting idea and the topic is of interest, but the machine learning methodology used by the authors lacks several key components.

There are several dozen commonly used machine learning algorithms, such as random forest, support vector machine, and neural network. The authors chose to use one particular algorithm - boosted regression tree, but did not justify why this algorithm is chosen and why other algorithms are not considered. According to the "no free lunch" theorem, no machine learning algorithm can always outperform other algorithms on every data set. The standard practice in machine learning is to try many different machine learning algorithms and pick the best one. This is not done in this study.

The paper does not mention for the approach of filling in missing values and the conclusions drawn from this approach to be valid, how accurate the machine learning model needs to be. Some of the accuracy values reported by the authors seem to be low. For example, lines 311-312 mentions that "BRT models achieved good accuracy both among cases (mean of percentage 312 of correct classifications 71% [95% CI: 43 - 100%]) and non-cases (74% [95% CI: 70 - 78%])". These are not high accuracies and should not be regarded as good accuracies. The readers are not assured that the analysis results done by filling in missing values are trustworthy.

The authors selected hyper-parameter values through grid search. Grid search is known to not perform well and unable to test many different values of a specific hyper-parameter. Actually for each hyper-parameter of boosted regression tree, the authors tested 3-5 values. This is too few and is unlikely to find near-to-optimal hyper-parameter values, whereas hyper-parameter values can have significant impact on the resulting machine learning model's accuracy. To address the issue/difficulty of selecting hyper-parameter values and machine learning algorithms, researchers have proposed various methods like automatic search and random search. The authors can check the following papers describing the state of the art and address this issue accordingly:

- Thornton C, Hutter F, Hoos HH, Leyton-Brown K. Auto-WEKA: combined selection and hyperparameter optimization of classification algorithms. Proc. KDD 2013:847-55.
- Komer B, Bergstra J, Eliasmith C. Hyperopt-sklearn: automatic hyperparameter configuration for scikit-learn. Proc. SciPy 2014:33-9.
- Zeng X, Luo G. Progressive sampling-based Bayesian optimization for efficient and automatic machine learning model selection. Health Inf Sci Syst 2017;5(1):2.
- Bergstra J, Bengio Y. Random search for hyper-parameter optimization. Journal of Machine Learning Research 2012;13:281-305.

Lines 126-129 mentions that "The optimal size of the training set and the optimal set of tuning parameters were determined with 127 reference to the out-of-sample sensitivity

(fraction of baseline seropositive subjects correctly 128 classified), specificity (fraction of baseline seronegative subjects correctly classified) and percentage 129 of predictions correctly classified overall". The optimal size and hyper-parameter values should be chosen based on a single criterion rather than multiple criteria. It is unclear how the authors are able to choose the optimal size and hyper-parameter values according to three different criteria simultaneously.

Line 130 mentions that "We then performed backward variable selection¹⁸". The authors did not mention why automatic feature selection is not used. In particular, tree-based machine learning algorithms do variable selection by default. Why this is not used here?

The paper is well written in general. There are several minor grammar issues that need to be fixed.

- In the sentence on lines 81-83 "Pooled analysis of the results obtained in CYD14 and CYD15 82 showed that among children of 9-16 years of age efficacy was 81.9% [95% CI: 67.2, 90.0] in 83 seropositive subjects and 52.5% [95% CI: 5.9, 76.1] seronegative subjects.", a comma should be added after "years of age".

- In the sentence on line 86 "Due to lack of power in the data caused by the relatively 86 small number of subjects tested at baseline", please add "a" before "lack of power".

Reviewer #3 (Remarks to the Author):

The work described in this manuscript is based on one Phase 2b and two Phase 3 trials of the Sanofi dengue vaccine. In the original studies only a small fraction of study subjects were characterized with regards to pre- and post-vaccination dengue serology, and the authors use the available serology data to explore how baseline serostatus and age influence vaccine efficacy. They impute the baseline dengue serostatus of 667 subjects with missing baseline serostatus but observed post dose 3 titres and then use the data from subjects with observed and imputed baseline dengue serostatus to infer the group-level baseline serostatus of the more than 30,000 subjects who had missing PD3 titres and missing baseline serostatus. The work is relevant to the ongoing controversy over the benefits and risks associated with the use of this dengue vaccine.

Reviewer's comments:

Line 34-37 (abstract): This does not appear to be a balanced summary of the available data. Serostatus at baseline clearly remains the dominant factor in determining efficacy. The higher frequency of dengue hospitalization in the youngest (likely seronegative) vaccinees versus controls should be addressed as well.

Line 67 and 83-85: The statement in Line 83-85 on efficacy by baseline serostatus does not provide the reader with the necessary information. When discussing the safety profile of the vaccine, the safety signal in inferred seronegative subjects needs to be addressed. In late 2017, WHO stated that "the subset of trial participants who were inferred to be seronegative at time of first vaccination had a significantly higher risk of more severe dengue and

hospitalizations from dengue compared to unvaccinated participants, regardless of age at time of vaccination. Beyond an initial protective period during the first two years, the risk was highest in year 3 following the first dose, declined in the following years but persisted over the trial follow up period of about 5 years after the first dose”.

http://www.who.int/immunization/diseases/dengue/q_and_a_dengue_vaccine_dengvaxia_use/en/

The most recent WHO SAGE recommendation to only vaccinate seropositive subjects should be mentioned and discussed. This would require that subjects be tested for dengue serostatus prior to vaccination.http://www.who.int/immunization/diseases/dengue/revised_SAGE_recommendations_dengue_vaccines_apr2018/en/

Line 68-70: The immune response to the vaccine post Dose 1 does not appear to be balanced. DENV4 behaves differently from the other 3 serotypes in that titers in seronegatives rise rapidly post dose 1 and do not rise much post Dose 2 or Dose 3. The findings in the authors’ 2015 publication (PMID: 26051515), especially the data in Tables S17 and S18 should be included in the introduction and be discussed. The data suggests to the reviewer that vaccination of baseline seronegative subjects acts like a primary DENV4 infection, which would explain the increased risk of hospitalization with a secondary non-DENV4 serotype.

Results:

Unfortunately I am not able to assess the mathematics underlying the modelling methodology. However, the use of post dose 3 titers (post dose 1 and post dose 2 titers are not available for these studies) to infer likely baseline serostatus is less robust than the use of post dose 1 titers would have been since with repeated exposure to vaccine the titers in baseline seronegatives, baseline monotypic seropositives and multitypic seropositives are in a narrower range than post dose 1 (see Dorigatti et al., 2015, PMID: 26051515, Tables S17 and S18).

In addition, repeated exposure induces cross-reactive antibodies and makes interpretation on serotype-specific antibody responses almost impossible. The outcome of the modelling exercise should therefore be treated with much caution and not be seen as “correcting” the actual findings of Phase 2b and Phase 3 studies. The actual findings stand as they are and the conclusions from this modeling exercise should be worded more cautiously.

Discussion:

The authors should discuss potential confounders that may have impacted the outcome of their analyses. In particular, the effect of flavivirus priming (via vaccination or infection) prior to vaccination needs to be discussed. Could the observed increase in efficacy with age in dengue seronegative subjects be due to prior dengue infection that is not detected in the PRNT assay? The likelihood of dengue infection increases with age and the likelihood of a long interval between dengue infection and baseline serology increases with age, therefore older subjects are more likely to have had a prior dengue infection yet test seronegative.

What percentage of subjects in the immune subset and the entire study population received

a flavivirus vaccine other than the investigational dengue vaccine, e.g. Japanese encephalitis vaccine or Yellow fever vaccine, and at what age? Was flavivirus vaccination considered in the modelling and if not, how could have it affected the outcome and interpretation?

Lines 281 & 282. The statement regarding mass vaccination campaigns needs to be corrected to indicate that The Philippines have stopped mass vaccination due to safety concerns.

Point-by-point response to referee's comments

Refining the characterization of the Sanofi Pasteur dengue vaccine's efficacy profile using machine learning

Dorigatti I, Donnelly CA, Laydon DJ, Small R, Jackson N, Coudeville L, Ferguson NM

Reviewer #1 (Remarks to the Author):

This research seeks to provide "a full characterization of how the efficacy of CYD-TDV varies by pre-exposure". Previous studies have not been able to do this due to the fact that only 12% of subjects in all trials had baseline serostatus data. In this paper optimised boosted regression trees (BRT) are used to impute the missing baseline serostatus data, allowing a more powerful analysis than was possible previously. Great care was taken to optimise the final BRT model and the relative importances for the predictor variables are useful.

However, further details are required in order to provide more confidence in the results relating to the imputation and the vaccine efficacy estimates. Ideally this evidence should be provided in the paper itself rather in what is a very long and probably not ideal set of supplementary materials. Figure S1-S9 are particularly difficult to read.

It is suggested that the authors provide detailed descriptive data regarding the number of (dengue) cases and non-cases for each age group. This should be done separately for the data for which baseline serostatus was known and where it was unknown. A comparison of these two data sets is also required in terms of other variables such as gender, age, country etc.

It is being assumed that one can impute baseline serostatus for 88% of the subjects from only 12% of the subjects, so some assurance is required as to the similarity of these two data sets in order to justify the whole process of baseline serostatus imputation. The relatively small number of (dengue) cases (155) among the 4251 subjects with baseline serostatus makes this descriptive analysis especially important. What is the percentage of dengue cases for the remaining data?

What percentage of these subjects are sero+ and sero- for each of the data sets, broken down by age group. Even powerful bootstrap methods have limited value when there are very small numbers of (dengue) cases.

We have now included a detailed descriptive analysis of the data in the main text and provided in Table R1 in this response a breakdown of the number of cases and non-cases by baseline serostatus and by all other available covariates (i.e. age, arm, country of enrolment, gender, PD3 titre availability against any serotype, serotype of infection (for the cases) and trial). In Table R1 we also included the percentage of cases that occurred in each stratum among subjects with known and unknown baseline serostatus and the p-values obtained from the Pearson's chi-squared test which was used to assess the significance of the difference in the number of cases observed in each stratum. Over all ages, 3.6% (155/4251) of the subjects with known baseline serostatus and 4% (1242/29527) of the subjects with unknown baseline serostatus were cases; the difference between these proportions is not statistically significant (p-value = 0.239).

Table R1 Descriptive summary of the number of cases and non-cases occurred in the phase-2b and phase-3 trials (CYD23, CYD14 and CYD15) during the active surveillance phase (up to 25 months after the first dose), stratified by baseline serostatus, age, arm, country of enrolment, gender, PD3 titre availability, serotype of infection (for cases) and trial. Sero+ stands for baseline seropositive and sero- stands for baseline seronegative. In each stratification the percentage of baseline seropositive subjects was calculated by dividing the number of seropositive subjects by the total number of seropositive and seronegative subjects; the percentages of cases were calculated by dividing the number of cases by the total number of cases and non-cases. The p-values were obtained using the Pearson's chi-squared test using the number of cases and non-cases with known and unknown baseline serostatus in each stratification. Significant (< 0.05) p-values are in bold.

		Known baseline serostatus							Unknown baseline serostatus			P-value chi-squared test (case vs knowledge of baseline serostatus)
		cases			non-cases			% cases	cases	non-cases	% cases	
		sero+	sero-	% sero+	sero+	sero-	% sero+	sero+ & sero-				
Age-group	All ages	90	65	58.1%	3,034	1,062	71.4%	3.6%	1,242	29,527	4.0%	0.239
	2-8 years	35	31	53%	700	483	56.0%	5.3%	405	6,073	6.3%	0.213
	9-11 years	28	20	58.3%	1,042	321	73.8%	3.4%	473	11,703	3.9%	0.412
	12-16 years	27	14	65.9%	1,292	258	81.2%	2.6%	364	11,751	3.0%	0.385
Arm	CYD-TDV	28	37	43.1%	2,079	695	73.2%	2.3%	579	19,943	2.8%	0.118
	Placebo	62	28	68.9%	955	367	67.6%	6.4%	663	9,584	6.5%	0.936
Country	Brazil	9	3	75.0%	210	78	68.2%	4.0%	107	3,137	3.3%	0.633
	Colombia	15	8	65.2%	829	61	90.8%	2.5%	250	8,555	2.8%	0.651
	Honduras	1	0	100%	257	37	87.1%	0.3%	112	2,379	4.5%	0.001
	Indonesia	11	8	55%	269	51	79.3%	5.6%	64	1,447	4.2%	0.339
	Mexico	4	3	57.1%	167	147	52.0%	2.2%	128	3,000	4.1%	0.126
	Malaysia	2	4	33.3%	139	152	46.8%	2.0%	24	1,075	2.2%	1
	Philippines	21	9	70.0%	447	117	75.3%	5.1%	265	2,611	9.2%	0.001
	Puerto Rico	2	4	33.3%	82	63	54.3%	4.0%	18	1,145	1.5%	0.076
	Thailand (CYD23)	7	6	53.8%	203	84	67.7%	4.3%	122	3,580	3.3%	0.429
	Thailand (CYD14)	13	9	59.1%	216	98	64.3%	6.5%	68	759	8.2%	0.396
Gender	Female	40	29	58%	1,559	538	72.0%	3.2%	627	15,078	4.0%	0.078
	Male	50	36	58.1%	1,475	524	70.7%	4.1%	615	14,449	4.1%	0.974
PD3 titres against any serotype	Known	79	48	62.2%	2,960	1,032	71.9%	3.1%	658	19	97.2%	< 0.001
	Unknown	11	17	39.3%	74	30	56.1%	21.2%	584	29,508	1.9%	< 0.001
Serotype	DENV1	27	26	50.9%	-	-	-	-	410	-	-	-
	DENV2	20	17	54.1%	-	-	-	-	370	-	-	-
	DENV3	21	9	70%	-	-	-	-	217	-	-	-
	DENV4	18	11	62.1%	-	-	-	-	201	-	-	-
	Unknown	4	2	66.7%	-	-	-	-	44	-	-	-
Trial	CYD23 (phase-2b)	7	6	53.8%	203	84	67.7%	4.3%	122	3,580	3.3%	0.429
	CYD14 (phase-3)	52	41	55.9%	1,286	592	65.2%	4.7%	505	7,731	6.1%	0.019
	CYD15 (phase-3)	31	18	63.3%	1,545	386	78.0%	2.5%	615	18,216	3.3%	0.066

By study design, all cases in the trials were retrospectively tested for dengue antibody levels but among the non-cases only those included in the immunogenicity subset (i.e. the subset of subjects with characterised PRNT titres at baseline and after each vaccine dose) had PD3 titres characterised. The study design thus explains the significant difference observed in the proportions of cases of cases with known and unknown PD3 titres between the subsets with known or unknown baseline serostatus (p-values < 0.001).

We also found significant differences in the number of cases and non-cases between subjects with known/unknown baseline serostatus in Honduras (p-value = 0.001), the Philippines (p-value = 0.001) and in the phase-3 CYD14 trial (p-value = 0.019). These differences are due to the lack of full randomisation in the assignment of a pre-defined number of subjects to the immunogenicity subsets in each site of the trials [1-3] (which were established according to the chronological time of subject enrolment in the trials).

In our analysis we used BRT to impute the baseline serostatus for 2.2% (677 out of 30,769) of the subjects with missing information, i.e. those with known PD3 titres. Imputation for the remaining subjects was conducted at the group-level. The differences observed between the cases and non-cases with known and unknown baseline serostatus in Honduras, the Philippines and CYD14 do not undermine the robustness of our analysis, given that the model used for individual-level imputation was purposely built using cases and non-cases over all trials and that case occurrence, as well as trial, were discarded from the final BRT model used for imputation given their limited contribution towards the predictive performance (see Table S2).

Summary of changes:

- In the paper, we have now included Table 1 (same as Table R1 shown above).
- In the Data section, we introduced the descriptive analysis presented in Table 1 as follows: Lines 134-140: *"Table 1 provides a descriptive summary of the number of cases and non-cases occurred in the phase-2b and phase-3 trials (CYD23, CYD14 and CYD15) during the active surveillance phase, stratified by baseline serostatus, age, arm, country of enrolment, gender, PD3 titre availability against any serotype, serotype of infection (for the cases) and trial. Additional statistics on the dataset are provided in section 1 of the Supplementary Material (SM) and further descriptive summaries of the total number of cases infected by serotype, age-group and baseline serostatus are provided in the SM, section 3.5.1."*
- In the Results section, we referred to the p-values reported in Table 1 as follows: Lines 199-208: *"Table 1 summarises the results of the Pearson's chi-squared test that was conducted to assess the significance of the difference between the frequencies of cases and non-cases with known and unknown baseline serostatus across multiple stratifications. Over all ages, 3.6% (155/4,251) of the subjects with known baseline serostatus and 4% (1,242/29,527) of the subjects with unknown baseline serostatus were cases; the difference between these proportions is not statistically significant (p-value 0.239). Among all other stratifications considered in Table 1, we found statistically significant differences in the number of cases and non-cases between subjects with known and unknown baseline serostatus in Honduras and in the Philippines (p-value 0.001), in the South East Asian (CYD14) phase-3 trial (p-value 0.019) and among subjects with known and unknown PD3 titres (p-values < 0.001)."*
- In the Discussion section, we discuss the results of the comparisons in Table 1 as follows: Lines 348-358: *"The statistically significant differences in the proportions of cases and non-cases between subjects with known and unknown baseline serostatus in Honduras, the Philippines (p-value = 0.001, see Table 1) and in the South East Asian (CYD14) phase-3 trial (p-value = 0.019, see Table 1) are likely due to the lack of full randomisation in the assignment of a pre-defined number of subjects to the immunogenicity subsets (i.e. the subset of subjects with known baseline serostatus) in each site of the trials^{15,16}, which were*

instead established according to the time of subject enrolment in the trials, in a chronological fashion. The statistically significant difference observed in the proportions of cases of cases with known and unknown PD3 titres between the subsets with known or unknown baseline serostatus (p-values < 0.001, see Table 1) was due to the fact that the trial design specified that PD3 blood samples were retrospectively tested for dengue antibody levels for all dengue cases plus all participants in the immunological subsets of the trials”

The methods section reports weighted linear regression analyses and Pearson Chi-Square tests for two or three age groups. I could not find details about these analyses in the supplementary materials. Using average efficacies rather than the subject data itself seems to be insufficient. On the other hand Pearson Chi-Squared tests, if used with subject data, would be unreliable with such large sample sizes. These analyses are critical in terms of the conclusions, so more convincing analyses and explanations are required for the vaccine efficacy results.

No detailed hypothesis testing results have been supplied. Instead there are 12 graphs with a somewhat confusing layout.

We have now included all details on the results obtained with weighted linear regression and the Pearson’s chi-squared test for two or three age-groups in SM sections 3.5 and 3.6. In these sections we have provided the estimates of the coefficients obtained from weighted linear regression, descriptive summaries of the aggregated data and the p-values obtained with the Pearson’s chi-squared test when comparing the vaccine efficacy estimates between the age-groups, for the analysis with and without imputation.

In addition, we further investigated the significance of the age-trend using a finer age-stratification into 2-years age-groups (2-3, 4-5, 6-7, 8-9, 10-11, 12-13 and 14-16 years), which provides a more detailed characterisation of the vaccine efficacy by age compared to the stratification used in the main analysis (2-8, 9-11, 12-16 years). Using a finer age-stratification into 2-years age-groups we still find a statistically significant age-trend among baseline seronegative subjects (Figure R1, included below), with an estimated 2.9% [95% CI: 0.4, 5.4%] (p-value = 0.03) increase in vaccine efficacy for each year-increase in age. This is consistent with the estimates obtained in our previous analyses and presented in the paper (i.e. 4.7% [95% CI: 0.05, 9.3%], p-value = 0.05 using imputation on the South East Asian trials using 2-5, 6-11 and 12-14 years age-groups and 4.6% [95% CI: 0.4, 8.9%], p-value = 0.04 without imputation using all trials and 2-8, 9-11 and 12-16 years age-groups). Consistently with our previous analyses, also using an age stratification into 2-years age-groups we found a non-significant age-trend among baseline seropositive subjects.

Finally, to clarify the layout of the estimates presented in Figure 1D-1F and address the lack of formal hypothesis testing in the main text, we have included a table (Table R2, see below) summarising the vaccine efficacy estimates by baseline serostatus and serotype obtained with imputation across the three main age-groups (2-16, 2-8 and 9-16 years old) and the results of the Pearson’s chi-squared test that we conducted to statistically compare the estimated vaccine efficacies against each serotype between baseline seronegative and baseline seropositive subjects.

Figure R1 Mean (solid line) and 95% confidence interval (dashed lines) of the age-trend in the estimated vaccine efficacy among baseline seronegative subjects, using age-groups 2-3, 4-5, 6-7, 8-9, 10-11, 12-13 and 14-16 years, with imputation of the baseline serostatus from 1,000 realisations of the final BRT model trained on 50% of non-cases and 75% of cases and using 10-fold cross-validation, $tc = 16$, $lr = 0.0005$, $bf = 0.75$.

Table R2 Mean and 95% CI of the vaccine efficacy estimates by baseline serostatus, serotype and age-group generated with imputation and significance of the statistical difference in vaccine efficacy between baseline seropositive and baseline seronegative subjects against each serotype in each age-group. The p-values were obtained using the Pearson's chi-squared test. Sero+ stands for seropositive; sero- stands for seronegative. Significant (<0.05) p-values and the corresponding estimates are in bold.

	All ages (2-16 year olds)			2-8 year olds			9-16 year olds		
	VE sero+	VE sero-	p-value	VE sero+	VE sero-	p-value	VE sero+	VE sero-	p-value
DENV1	0.59 (0.38, 0.74)	0.45 (0.22, 0.64)	0.309	0.51 (-0.12, 0.81)	0.36 (-0.10, 0.66)	0.624	0.62 (0.42, 0.77)	0.52 (0.26, 0.70)	0.469
DENV2	0.54 (0.33, 0.73)	-0.08 (-0.70, 0.38)	0.029	0.34 (-0.17, 0.69)	-0.01 (-0.99, 0.56)	0.426	0.63 (0.44, 0.79)	-0.18 (-1.41, 0.36)	0.085
DENV3	0.74 (0.61, 0.84)	0.67 (0.38, 0.90)	0.630	0.68 (0.21, 0.95)	0.73 (0.10, 1.00)	0.885	0.75 (0.62, 0.85)	0.64 (0.27, 0.89)	0.485
DENV4	0.88 (0.80, 0.95)	0.50 (0.12, 0.72)	0.019	0.77 (0.37, 0.97)	0.22 (-1.28, 0.356)	0.379	0.91 (0.84, 0.97)	0.59 (0.11, 0.81)	0.070

Summary of changes:

- In SM section 3.5 (“Association between age and the serotype of infection by pre-exposure”) we have included an entire sub-section (SM section 3.5.1) reporting the results obtained without imputation, including the case counts by age-group and serotype (Tables S3-S6) and the p-values obtained from the Fisher’s exact test. The results obtained with imputation are now presented in SM section 3.5.2, where we have included a new figure showing the p-value distributions obtained with imputation.
- In the SM, we have added a new section (section 3.6, “Age-trends in vaccine efficacy”) reporting the estimates obtained with weighted linear regression and the p-values obtained with the Pearson’s chi-squared test on the data without imputation (SM section 3.6.1) and with imputation (SM section 3.6.2). We have included the estimates of the coefficients obtained from weighted linear regression using 2-year age-groups and also visual representations of the linear trends in the vaccine efficacy estimates.
- In the Results section, we have commented on the robustness of the estimated age-trend among baseline seronegative subjects as follows:

Lines 281-286: *“The significance and consistency of the age-trend in vaccine efficacy among baseline seronegative subjects was further confirmed by the results of weighted linear regression on the vaccine efficacy estimates obtained with imputation using all trials and a finer age-stratification into 2-year age-groups (i.e. 2-3, 4-5, 6-7, 8-9, 10-11, 12-13 and 14-16 years), where we found 2.9% [95% CI: 0.4, 5.4%] (p-value = 0.03) increase in vaccine efficacy for each year-increase in age (see SM section 3.6.2 for details).”*

- In the Results section, we have added Table 2 (same as Table R2 shown above).
- In the Results section, we introduced the results shown in Table 2 as follows:
Lines 244-247: *“Among all study subjects (2-16 years), we find significantly different vaccine efficacies against DENV2 and DENV4 between baseline seronegative and baseline seropositive subjects (Table 2). However, these differences were not statistically significant when calculated among children 2-8 and 9-16 years old separately.”*

A lot of work has gone into this paper. It is hoped that the above feedback will make it possible for future readers to better appreciate its value.

Thank you for the positive and constructive feedback. We believe that the additional information provided has greatly improved the paper, to the benefit of future readers.

Reviewer #2 (Remarks to the Author):

This reviewer gives a reject to this paper.

This paper describes the approach of building a machine learning model to fill in missing values to better estimate the efficacy of CYD-TDV, a dengue vaccine. This is an interesting idea and the topic is of interest, but the machine learning methodology used by the authors lacks several key components.

There are several dozen commonly used machine learning algorithms, such as random forest, support vector machine, and neural network. The authors chose to use one particular algorithm - boosted regression tree, but did not justify why this algorithm is chosen and why other algorithms are not considered. According to the “no free lunch” theorem, no machine learning algorithm can always outperform other algorithms on every data set. The standard practice in machine learning is to try many different machine learning algorithms and pick the best one. This is not done in this study.

We chose boosted regression trees (BRT) for data imputation for multiple reasons, including (i) its increasing application and success in similar statistical epidemiology problems [4-9] (ii) its ease of application and accessibility [10], (iii) its excellent performance observed in several comparative analyses applied to diverse datasets, including a comprehensive comparison conducted on a spatial classification problem [11], and (iv) its ability to incorporate incomplete or missing data in the predictors (as opposed to other algorithms that require complete data in the predictor variables both for training and prediction). This latter was an important consideration for our dataset, given that 18% (28 out of 155) of the cases and 2.5% (106 out of 4096) of the non-cases with observed baseline serostatus had incomplete data on at least one of the other predictors. In this regard, we preferred not to utilise automatic imputation techniques to fill in for missing data in the predictors (e.g. with the average observed across the dataset or other methods available in some machine learning algorithms).

However, we agree that it is interesting to compare the performance of the algorithm that we have chosen (BRT) with other common alternatives such as generalised linear models (GLM), random forest (RF), support vector machine (SVM) and neural network (NN), as suggested by the reviewer.

Figure R2 shows the accuracy measures obtained overall and among cases and non-cases separately using five well-established machine learning algorithms (BRT, GLM, RF, SVM and NN) and all available predictors, where the tuning parameters of each method were established by random search (for details see SM section 5). In the analysis presented in the paper using BRT we optimised two distinct cut-off probability thresholds used for classification, one for cases and one for non-cases – this allowed us to balance sensitivity and specificity among cases, as opposed to using a single cut-off threshold, which resulted in an average 14% lower sensitivity than specificity for the cases (see SM section 2.2). For consistency, in this analysis we optimised two distinct cut-off probability thresholds (one for cases and one for non-cases) for all algorithms, except SVM which by construction is built on the concept of distance and lacks a probabilistic implementation (although probabilistic interpretations have been proposed [12-13]).

The results of this comparative analysis suggest that BRT, GLM, NN and RF performed similarly in terms of sensitivity, specificity and percentage of predictions correctly classified overall and among non-cases, with RF showing a slightly higher predictive performance among cases (77%) as compared to BRT (74.5%), NN (72%), and GLM (68%). SVM produced higher sensitivities overall (90%) and among non-cases (90%) and higher specificity (86%) among cases than BRT, GLM, NN and RF but lower specificity (83%) overall and among non-cases and lower sensitivity (65%) among cases than BRT, GLM, NN and RF.

We therefore assessed the impact of the small increase in predictive performance observed among cases using RF compared with BRT (77% versus 74.5%) on the vaccine efficacy estimates obtained with imputation. For consistency with the estimates obtained using BRT, we first conducted automatic feature selection on the RF model and then used the final RF model (i.e. with only the important variables included) to generate the vaccine efficacy estimates with bootstrapping (all details are given in SM section 5).

Figure R3 shows a comparison of the accuracy and vaccine efficacy estimates obtained with imputation using the final BRT and RF models. We find that BRT and RF perform equally in terms of all out-of-sample accuracy measures, overall and among both cases and non-cases (Figure R3A). Figures R3B-3F show that the vaccine efficacy estimates obtained with imputation through RF are consistent with vaccine efficacy estimates obtained with imputation using BRT. With the final RF model and using 2-8, 9-11 and 12-16 years age-groups we found a significant age-trend among baseline seronegative subjects giving 3.8% [95% CI: 0.4, 7.2%] (p-value = 0.04) increase in vaccine efficacy for each year-increase in age. The age-trends in vaccine efficacy estimated among baseline seronegative subjects using RF are consistent with the estimates obtained with BRT (SM section 3.6). These results provide further evidence of the robustness of the estimates presented in the main analysis.

In conclusion, we have compared the performance of five different machine learning algorithms on our classification problem and demonstrated that the use of BRT and RF (the two optimal classification algorithms) for imputation produce statistically equivalent vaccine efficacy estimates, thus providing evidence that the results produced in our analysis are robust to the machine algorithm chosen. We have now included all details on the comparative analysis conducted between the different machine learning algorithms in SM section 5.

Figure R2 Out-of-sample sensitivity (proportion of baseline seropositive subjects correctly classified), specificity (proportion of baseline seronegative subjects correctly classified) and proportion of correct classifications among cases, non-cases and overall using BRT, GLM, NN, RF and SVM models. Each model was parametrised using the respective optimal hyper-parameterisation found using random hyper-parameter search and the optimal size of the training sets. Each method was built on 100 randomly drawn training and validation sets, using the algorithm described in SM section 2.2.

Figure R3 Mean and 95% CI of the vaccine efficacy estimates generated with imputed data for baseline seronegative (sero-) and baseline seropositive (sero+) subjects separately using BRT and RF. Estimates were obtained from 1,000 realisations of the final BRT and RF models. A) sensitivity (sens), specificity (spec) and proportion of correct classifications (pcc) among cases, non-cases and overall; B) vaccine efficacy estimates for baseline seropositive and baseline seronegative subjects

separately; C) vaccine efficacy estimates for baseline seropositive and baseline seronegative subjects by age using 2-8, 9-11, 12-16 and 9-16 years age-groups ; D) vaccine efficacy estimates by serotype for baseline seropositive and baseline seronegative subjects of all ages (2-16 years); E) vaccine efficacy estimates by serotype for baseline seropositive and baseline seronegative subjects 2-8 years old; F) vaccine efficacy estimates by serotype for baseline seropositive and baseline seronegative subjects 9-16 years old.

Summary of changes:

- In SM section 5, we present the results of the comparative analysis conducted to assess the predictive performance of five different machine learning algorithms (BRT GLM, NN, RF and SVM). In SM section 5 we have included details on (i) the random search that was conducted to establish the optimal hyper-parameterisation of each algorithm, (ii) automatic feature selection for RF and (iii) the vaccine efficacy estimates with imputation obtained using the final BRT and RF models.
- In the Discussion, we refer to the comparative analysis presented in SM section 5 as follows: Lines 362-365: *“Analysis of the accuracy achieved by BRT compared to other commonly adopted machine learning algorithms (including generalised linear models, random forest and neural networks) is presented in SM section 5 and shows the optimal predictive performance of BRT on the data analysed in this study.”*

The paper does not mention for the approach of filling in missing values and the conclusions drawn from this approach to be valid, how accurate the machine learning model needs to be. Some of the accuracy values reported by the authors seem to be low. For example, lines 311-312 mentions that “BRT models achieved good accuracy both among cases (mean of percentage of correct classifications 71% [95% CI: 43 – 100%]) and non-cases (74% [95% CI: 70 – 78%])”. These are not high accuracies and should not be regarded as good accuracies. The readers are not assured that the analysis results done by filling in missing values are trustworthy.

As discussed in our previous response, one of the appealing properties of BRT compared to several other machine learning algorithms such as RF for instance, is its ability to incorporate missing data in the predictors. For this reason, we did not have to recur to any automatic imputation to fill in for missing data in the predictors in our analysis (e.g. with the average observed across the dataset or other automatic methods available in some machine learning algorithms).

Using the final BRT model and a binary classification of the baseline serostatus (seropositive, seronegative) we achieved a mean percentage of correct classifications of 86% [95% CI: 84% - 88%] overall and among non-cases and 75% [95% CI: 50 – 100%] among cases. Using a multinomial classification of the baseline serostatus (seronegative, monotypic, multitypic) we obtained the accuracies that the reviewer is referring to, i.e. a mean percentage of correct classifications of 74% [95% CI: 70 – 78%] overall and among non-cases and of 71% [95% CI: 43 – 100%] among cases. The lower accuracies obtained using a multinomial classification of the baseline serostatus compared to a binary classification is driven by a lack of power when including multiple stratifications. We appreciate that these accuracies may not be considered ‘good’ in some applications and we have reworded the text accordingly. At the same time, we believe that it is interesting and important to report the results obtained using a more refined classification, which gives further insight into the effect of baseline pre-exposure on vaccine efficacy.

Importantly, imputation in clinical trials is traditionally conducted within a parametric framework (using the ‘multiple imputation’ approach that is based on regression and Rubin’s rule), which makes the application of machine learning to the analysis of clinical trial data particularly relevant. In the analysis of clinical trial data, it is not common practice to use pre-defined or heuristic threshold accuracies. The performance of imputation in fact depends on the quality and quantity of data available, beyond on the goodness of the model.

In this study, we evaluated the model predictive performance using specific individual-level measures (i.e. out-of-sample sensitivity, specificity and proportion of predictions correctly classified) that were calibrated to avoid imputation introducing bias into the results (i.e. balanced sensitivity and specificity). Evidence that imputation by BRT does not bias the vaccine efficacy estimates is provided by the consistency of the seroprevalence with and without imputation across multiple stratifications (Figures S12 and S31-33, S35-37). Ultimately, we believe that a model that does not bias the data and correctly predicts the baseline serostatus on average 75% [95% CI: 50 – 100%] of the times among cases and 86% [95% CI: 84% - 88%] of the times among non-cases can be used to fill in the missing baseline serostatus on 2.2% (677 out of 30,769) of the subjects in the trial. Should the statistics presented in the paper appear insufficient, we request that the reviewer identifies specific metrics that they would need to be reassured that the results are informative.

Summary of changes:

- In the Discussion, we have reworded the presentation of the predictive performance obtained using the multinomial classification as follows:
Lines 417-421: “Using a more refined partitioning of baseline seropositive subjects into monotypic or multitypic PRNT50 profiles we found that BRT models achieved lower accuracy both among cases (mean of percentage of correct classifications 71% [95% CI: 43 – 100%]) and non-cases (74% [95% CI: 70 – 78%]).”

The authors selected hyper-parameter values through grid search. Grid search is known to not perform well and unable to test many different values of a specific hyper-parameter. Actually for each hyper-parameter of boosted regression tree, the authors tested 3-5 values. This is too few and is unlikely to find near-to-optimal hyper-parameter values, whereas hyper-parameter values can have significant impact on the resulting machine learning model’s accuracy. To address the issue/difficulty of selecting hyper-parameter values and machine learning algorithms, researchers have proposed various methods like automatic search and random search. The authors can check the following papers describing the state of the art and address this issue accordingly:

- Thornton C, Hutter F, Hoos HH, Leyton-Brown K. Auto-WEKA: combined selection and hyperparameter optimization of classification algorithms. Proc. KDD 2013:847-55.
- Komer B, Bergstra J, Eliasmith C. Hyperopt-sklearn: automatic hyperparameter configuration for scikit-learn. Proc. SciPy 2014:33-9.
- Zeng X, Luo G. Progressive sampling-based Bayesian optimization for efficient and automatic machine learning model selection. Health Inf Sci Syst 2017;5(1):2.
- Bergstra J, Bengio Y. Random search for hyper-parameter optimization. Journal of Machine Learning Research 2012;13:281-305.

In the main analysis we optimised the hyper-parameterisation using a grid search that comprised 540 different parameterisations resulting from the combination of 3 to 5 values for 5 hyper-parameters (tree complexity, learning rate, bag fraction, proportion of cases and proportion of non-cases with complete data included in the training set). From this analysis we found that the optimal hyper-parameterisation was given by tree complexity = 16, learning rate = 0.0005, bag fraction = 0.75 and training sets composed by 75% of cases and 50% of non-cases with complete information on the baseline serostatus. The BRT models obtained with this parameterisation had on average 15,311 trees.

We agree with the reviewer that algorithms based on random search may be better suited for high-dimensional optimisation in general and we explored the effect that the choice of the algorithm used to select the hyper-parameterisation (i.e. grid versus random search) had on the out-of-sample sensitivity, specificity and proportion of predictions correctly classified among cases and non-cases. We implemented random hyper-parameter search by drawing 1,000 random combinations of tree

complexity (integer distribution between 2 and 50), learning rate (uniform distribution between 0.0005 and 0.01), bag fraction (uniform distribution between 0.5 and 1) and numbers of cases and non-cases with complete information on the baseline serostatus included in the training set (integer distribution between 39 and 116 for cases and between 1,024 and 3,072 for non-cases). The optimal parameterisation obtained in the random search had tree complexity = 43, learning rate = 0.002, bag fraction = 0.71 and training sets composed by 59% (91/155) of cases and 66% (2,729/4,096) of non-cases with complete information on the baseline serostatus. The BRT models obtained with this parameterisation had on average 3,538 trees.

Figure R4 shows the out-of-sample sensitivity, specificity and proportion of predictions correctly classified among cases and non-cases obtained with the optimal hyper-parameterisations from grid and random search. Figure R4 shows that the performance of the optimal parameterisations obtained by grid and random search are equivalent. This relative robustness of BRT's performance on the choice of the hyper-parameterisation is due to the fact BRT optimises the number of trees grown according to the hyper-parameterisation used (this is markedly different from random forest for instance, where the total number of trees grown is given as a parameter). Random search identified a hyper-parametrisation that achieved the same accuracy obtained with grid search but using less computational time (i.e. requiring a smaller number of trees to be grown).

Figure R4 Out-of-sample sensitivity, specificity and proportion of correct classifications (probability of correctly classifying the baseline serostatus) among cases and non-cases (mean and 95% CI) using the optimal parameterisation found using grid search ($tc = 16$, $lr = 0.0005$, $bf = 0.75$ and training sets composed by 75% of cases and 50% of non-cases with complete information on the baseline serostatus) and random search ($tc = 43$, $lr = 0.002$, $bf = 0.71$ and training sets composed by 59% (91/155) of cases and 66% (2,729/4,096) of non-cases with complete information on the baseline serostatus).

Summary of changes:

- In the Results section, we present the results of the comparative analysis of BRT performance using grid and random search as follows:
Lines 217-220: *“Sensitivity analysis on the performance of a random search to identify the optimal parameterisation showed that the accuracies achieved with BRT are robust to the choice of the searching algorithm adopted (i.e. random versus grid search) and the specific optimal parameters used (details are given in SM, section 3.1).”*

- In SM section 3.1, we have included the results obtained using random hyper-parameter search and a discussion of the relevance of random versus grid search in the context of BRT.

Lines 126-129 mentions that “The optimal size of the training set and the optimal set of tuning parameters were determined with reference to the out-of-sample sensitivity (fraction of baseline seropositive subjects correctly classified), specificity (fraction of baseline seronegative subjects correctly classified) and percentage of predictions correctly classified overall”. The optimal size and hyper-parameter values should be chosen based on a single criterion rather than multiple criteria. It is unclear how the authors are able to choose the optimal size and hyper-parameter values according to three different criteria simultaneously.

In our modelling framework we optimised two separate cut-off thresholds used for classification, one for cases and one for non-cases (separately) so that imputation returned the same sensitivities and specificities [14]. This choice was made to avoid imputations introducing bias into the data. This implies that the proportion of predictions correctly classified for cases and non-cases (separately) is also equal to the sensitivity/specificity of the cases and non-cases (separately).

We used the following rationale to identify the optimal parameterisation. First, we identified the maximum sensitivity/specificity/percentage of predictions correctly classified among cases. Then, we selected all parameterisations returning sensitivity/specificity/percentage of predictions correctly classified statistically equivalent (i.e. with overlapping confidence interval) to the selected maximum. Finally, among the parameterisations selected in the previous step, we selected the hyper-parameterisation with the highest sensitivity/specificity/percentage of predictions correctly classified among non-cases. We preferred to prioritise the maximisation of the accuracy measures among cases because imputation was mainly conducted on cases due to the trial design.

Specifically, the optimal hyper-parameterisation adopted in the final BRT model (i.e. tree complexity (tc) = 16, learning rate (lr) = 0.0005, bag fraction (bf) = 0.75 and training sets composed by 75% of cases and 50% of non-cases with complete information on the baseline serostatus) was selected using the following algorithm:

1. Calculate the highest lower bound of the confidence interval of sensitivity/specificity /proportion predictions correctly classified among cases.
2. Select the parameterisations that produce statistically equivalent sensitivity/specificity/ proportion predictions correctly classified among cases. This is obtained by selecting the parameterisations that have the upper bound of the confidence interval of sensitivity/specificity/proportion predictions correctly classified among cases larger or equal than the indicator obtained in step 1.
3. Among the parameterisations selected in step 2, choose the one giving the highest upper bound of the confidence interval of the sensitivity/specificity/proportion of predictions correctly classified among non-cases.

Summary of changes:

- In SM section 3.1 we clarified the rationale used to select the optimal parameterisation and specified the steps used to select the optimal hyper-parameterisation, as shown above.

Line 130 mentions that “We then performed backward variable selection”. The authors did not mention why automatic feature selection is not used. In particular, tree-based machine learning algorithms do variable selection by default. Why this is not used here?

We referred to “backward variable selection” in the sense of “variable elimination” and used indeed automatic feature selection, as detailed in SM section 2.3 and in the documentation of the brt

package¹. We recognise that the word “backward” was used improperly and have reworded the text accordingly.

Summary of changes:

- In the Methods section, we corrected our claim as follows:
Line 161-163: *“We then performed variable elimination using automatic feature selection to drop the variables that gave no evidence of improving the predictive performance of the model²³⁻²⁵ (SM, section 2.3) and obtained a final model which was used to impute the baseline dengue serostatus of the 677 subjects with missing baseline serostatus but observed PD3 PRNT50 titres.”*
- Lines 221–223: *“Automatic feature selection indicated that trial and case occurrence could be dropped from the predictors with minimal (< 0.17%) changes in the out-of-sample prediction accuracy (SM, section 3.2).”*

The paper is well written in general. There are several minor grammar issues that need to be fixed.

- In the sentence on lines 81-83 “Pooled analysis of the results obtained in CYD14 and CYD15 82 showed that among children of 9-16 years of age efficacy was 81.9% [95% CI: 67.2, 90.0] in 83 seropositive subjects and 52.5% [95% CI: 5.9, 76.1] seronegative subjects.”, a comma should be added after “years of age”.

- In the sentence on line 86 “Due to lack of power in the data caused by the relatively small number of subjects tested at baseline”, please add “a” before “lack of power”.

Thank you for the suggestions, we have now revised the text and fixed all minor grammatical issues.

Summary of changes:

- Lines 90-92: *“Pooled analysis of the results obtained in CYD14 and CYD15 82 showed that among children of 9-16 years of age, efficacy was 81.9% [95% CI: 67.2, 90.0] in 83 seropositive subjects and 52.5% [95% CI: 5.9, 76.1] in seronegative subjects.”*
- Lines 107-110: *“Due to a lack of power in the phase-2b and phase-3 trial data caused by the relatively small number of subjects tested at baseline, a full characterization of how the efficacy of CYD-TDV varies by pre-exposure for further stratifications is lacking and there are uncertainties around existing efficacy estimates for baseline seronegative and baseline seropositive subjects.”*

Reviewer #3 (Remarks to the Author):

The work described in this manuscript is based on one Phase 2b and two Phase 3 trials of the Sanofi dengue vaccine. In the original studies only a small fraction of study subjects were characterized with regards to pre- and post-vaccination dengue serology, and the authors use the available serology data to explore how baseline serostatus and age influence vaccine efficacy. They impute the baseline dengue serostatus of 667 subjects with missing baseline serostatus but observed post dose 3 titres and then use the data from subjects with observed and imputed baseline dengue serostatus to infer the group-level baseline serostatus of the more than 30,000 subjects who had missing PD3 titres and missing baseline serostatus. The work is relevant to the ongoing controversy over the benefits and risks associated with the use of this dengue vaccine.

Reviewer’s comments:

Line 34-37 (abstract): This does not appear to be a balanced summary of the available data. Serostatus at baseline clearly remains the dominant factor in determining efficacy. The higher frequency of dengue hospitalization in the youngest (likely seronegative) vaccinees versus controls should be addressed as well.

We agree that it is important to present this analysis in the light of the higher risk of hospitalisation observed in year 3. This gives us the opportunity to stress the importance of gaining insight into the relationship between vaccine efficacy, age and prior dengue exposure, which is a key outcome of the study. We have now provided an improved summary of the available data in the abstract.

Summary of changes:

- In the abstract, we have added the following sentence:
Lines 30-35: *“Because age and dengue exposure are highly correlated in dengue endemic settings, refined insight into how the efficacy of CYD-TDV varies by serostatus and age is essential to better understand the increased risk of hospitalisation observed among vaccinated individuals younger than 9 years of age during year 3 of the long-term follow-up phase and for the development of safe and effective vaccination strategies.”*

Line 67 and 83-85: The statement in Line 83-85 on efficacy by baseline serostatus does not provide the reader with the necessary information. When discussing the safety profile of the vaccine, the safety signal in inferred seronegative subjects needs to be addressed. In late 2017, WHO stated that “the subset of trial participants who were inferred to be seronegative at time of first vaccination had a significantly higher risk of more severe dengue and hospitalizations from dengue compared to unvaccinated participants, regardless of age at time of vaccination. Beyond an initial protective period during the first two years, the risk was highest in year 3 following the first dose, declined in the following years but persisted over the trial follow up period of about 5 years after the first dose”.

http://www.who.int/immunization/diseases/dengue/q_and_a_dengue_vaccine_dengvaxia_use/en/

The most recent WHO SAGE recommendation to only vaccinate seropositive subjects should be mentioned and discussed. This would require that subjects be tested for dengue serostatus prior to vaccination.

http://www.who.int/immunization/diseases/dengue/revised_SAGE_recommendations_dengue_vaccines_apr2018/en/

We agree that a more comprehensive discussion of the safety results, beyond the efficacy results, published in the literature is needed to provide a full picture of the implications of the dependence of vaccine efficacy on the baseline serostatus. This is now extensively discussed in the introduction, together with the latest WHO recommendations to vaccinate only baseline seropositive subjects.

Summary of changes:

- In the Introduction, we have now included the latest WHO recommendations and extensively discussed CYD-TDV safety concerns as follows:
Lines 92-106: *“The increase in the relative risk of hospitalisation for virologically confirmed dengue in the vaccine group compared with the control group observed in the Southeast Asian trial (CYD14) among children 2-5 years old (relative risk 7.45% [95% CI: 1.15 – 313.80]) and younger than 9 years (relative risk 1.58 [95% CI: 0.61-4.83]) during year 3 of the long-term follow-up phase of surveillance (24 – 36 months post first vaccination)¹⁹, together with the less favourable efficacy results observed in young children^{17,18}, were key elements for choosing to license CYD-TDV among children of 9 years of age or older. New analysis of long-term data recently announced by Sanofi Pasteur^{20,21} confirmed the differences in CYD-TDV performance according to baseline serostatus and resulted in a change in the vaccination recommendations both by Sanofi Pasteur and the World Health Organization (WHO) – with CYD-TDV now only being recommended for subjects exposed to dengue prior to the time of vaccination within the indicated age range²². The higher risk of severe dengue and dengue-*

related hospitalization in baseline seronegative subjects up to 5 years post first vaccination reported by the WHO²³ indicate that there is a clear need for a better understanding of the role of age-specific effects, independent of serostatus."

Line 68-70: The immune response to the vaccine post Dose 1 does not appear to be balanced. DENV4 behaves differently from the other 3 serotypes in that titers in seronegatives rise rapidly post dose 1 and do not rise much post Dose 2 or Dose 3. The findings in the authors' 2015 publication (PMID: 26051515), especially the data in Tables S17 and S18 should be included in the introduction and be discussed. The data suggests to the reviewer that vaccination of baseline seronegative subjects acts like a primary DENV4 infection, which would explain the increased risk of hospitalization with a secondary non-DENV4 serotype.

We thank the reviewer for their inputs in the interpretation of the immunogenicity data published in Dorigatti et al [15] and agree that vaccination in baseline seronegative subjects appears to act like a primary dengue infection (as discussed in Ferguson et al [16]) and that it is reasonable to extrapolate from these results that CYD-TDV likely acts as a primary DENV4 infection. This hypothesis is consistent with the high vaccine efficacy estimates against DENV4 observed in the phase-2b and phase-3 trials and also with the increased risk of hospitalisation observed in vaccinated baseline seronegative subjects in the long term follow-up phase [17]. On the other hand, due to the absence of correlates of protection and the exclusive availability of immunogenicity data, this hypothesis could not be validated in Dorigatti et al [15]. As suggested, we have now included in the text a discussion of the immunogenicity results, in the light of the results available at the time.

Summary of changes:

- In the Introduction, we removed our statement on the induction of a 'balanced' antibody response and added a discussion of the implications of the results obtained from the immunogenicity analysis as follows:
Lines 67-77: *"Several trials have demonstrated the safe reactogenicity¹⁰ and good immunogenicity profile of the vaccine against all serotypes.^{11,12} A detailed analyses of multiple phase-2 trials of CYD-TDV revealed the fundamental role of dengue exposure prior to vaccination (herein referred to as baseline serostatus) on the vaccine immunogenicity and the ability of CYD-TDV to elicit a strong DENV4 antibody response since the first vaccine dose, that was comparable to the immunity observed upon natural infections in all subjects, including those with no evidence of dengue exposure before vaccination. While the vaccine-induced antibody titres against DENV1-3 appeared lower than the respective antibody response elicited upon natural infection¹², the absence of correlates of protection implied that no conclusion on the expected vaccine efficacy could be drawn from the analysis of the immunogenicity data alone."*

Results:

Unfortunately I am not able to assess the mathematics underlying the modelling methodology. However, the use of post dose 3 titers (post dose 1 and post dose 2 titers are not available for these studies) to infer likely baseline serostatus is less robust than the use of post dose 1 titers would have been since with repeated exposure to vaccine the titers in baseline seronegatives, baseline monotypic seropositives and multitypic seropositives are in a narrower range than post dose 1 (see Dorigatti et al., 2015, PMID: 26051515, Tables S17 and S18). In addition, repeated exposure induces cross-reactive antibodies and makes interpretation on serotype-specific antibody responses almost impossible. The outcome of the modelling exercise should therefore be treated with much caution and not be seen as "correcting" the actual findings of Phase 2b and Phase 3 studies. The actual findings stand as they are and the conclusions from this modeling exercise should be worded more cautiously.

We agree that imputation can be used to fill in for missing information on the baseline serostatus to increase the statistical power of the analysis and that this technique is not meant to correct the data. We have now more cautiously worded the conclusions of our study.

Summary of changes:

- In the Discussion, we more cautiously present the conclusions of this study as follows:
Lines 369-374: *“The increased precision of the vaccine efficacy estimates obtained with imputation suggests significantly positive efficacy among baseline seronegative children (i) 9-16 years when pooling serotypes, (ii) 9-16 years against DENV1, DENV3 and DENV4, (iii) against DENV1, DENV3 and DENV4 over all ages (Figure 1 and S13), (iv) in both CYD14 and CYD15 and (v) in Brazil, Colombia, Malaysia and the Philippines (Figures S12 and S14), where the efficacy estimates in the absence of imputation were not significantly positive.”*
Lines 378-381: *“However, our estimates obtained with imputation suggest non-significant efficacy against DENV2 among seronegative subjects using a variety of age-stratifications (2-16 (all ages), 2-8 and 9-16 years) and among seropositive subjects in the 2-8 years age-group.”*

Discussion:

The authors should discuss potential confounders that may have impacted the outcome of their analyses. In particular, the effect of flavivirus priming (via vaccination or infection) prior to vaccination needs to be discussed. Could the observed increase in efficacy with age in dengue seronegative subjects be due to prior dengue infection that is not detected in the PRNT assay? The likelihood of dengue infection increases with age and the likelihood of a long interval between dengue infection and baseline serology increases with age, therefore older subjects are more likely to have had a prior dengue infection yet test seronegative.

What percentage of subjects in the immune subset and the entire study population received a flavivirus vaccine other than the investigational dengue vaccine, e.g. Japanese encephalitis vaccine or Yellow fever vaccine, and at what age? Was flavivirus vaccination considered in the modelling and if not, how could have it affected the outcome and interpretation?

It is possible that, due to antibody decay below the lower limit of detection, the PRNT assay could miss to identify past dengue infections. However, in the absence of well characterised estimates of antibody decay following natural exposure or vaccination in different transmission settings, it is hard to assess the extent to which the seroprevalence estimates obtained with PRNT may underestimate the actual burden of dengue infection. Our opinion is that, in high-transmission settings such as those selected for the CYD-TDV trials, where on average primary infections occur early in life and antibody boosting due to post-primary infections are also frequent among young children, it is unlikely that antibody decay below detectable levels occurs in large proportions of the study population in relatively short time intervals (i.e. by 14 years of age in South East Asia and by 16 years of age in Latin America).

Unfortunately, in this study we did not have information on the baseline Yellow Fever (YF) or Japanese Encephalitis (JE) serostatus of the subjects (either due to vaccination or natural exposure) nor individual-level information on whether the subjects were vaccinated against YF or JE. For these reasons, we could not include the YF/JE baseline vaccination status/serostatus as predictors of the baseline dengue serostatus. The analysis conducted in Dorigatti et al [15] suggests that pre-existing immunity to JEV could induce a broader and stronger response to CYD-TDV vaccination. Similarly, vaccination with CYD-TDV in YFV pre-immune individuals could recall cellular responses against the YFV backbone strain, which could assist through a bystander effect on specific responses against the envelope proteins. This implies that the increased vaccine efficacy estimated in this study among older children could be indeed linked to the higher probability of JE/YF exposure (either due to natural infection or vaccination) with age. The similarity in the PD3 dengue titres observed between

(i) subjects testing seropositive to dengue and seronegative to JE at baseline and (ii) subjects testing seronegative to dengue and seropositive to JE at baseline (see PD3 titres in green colour in row 3 of Figure S4 in Dorigatti et al [15]) implies that information on the baseline JE vaccination/serostatus could potentially improve the predictive performance of the model developed in this analysis. The same could be conjectured in relation to the baseline YF vaccination/serostatus.

In the paper, we have now included a discussion of the potential role of JE/YF exposure on the study outcomes, including the model performance and the estimated age-trend in vaccine efficacy.

Summary of changes:

- In the Discussion, we have added a discussion of the potential effect of flavivirus priming on the outcomes of this study as follows:
Lines 391-402: *“Age dependence in efficacy could be due to maturation of the immune system^{37, 38}, age dependence in dengue infection severity³⁹ or a potential role of Japanese Encephalitis (JE) or Yellow Fever (YF) virus exposure, either due to natural infection or vaccination. This latter hypothesis is consistent with the analysis conducted in Dorigatti et al¹², where pre-existing immunity to JE was shown to induce a broader and stronger response to CYD-TDV vaccination. Similarly, vaccination with CYD-TDV in YF pre-immune individuals could recall cellular responses against the YF backbone strain, which could assist through a bystander effect on specific responses against the envelope proteins. Unfortunately, these assumptions could not be further validated due to the unavailability of information on pre-existing JE/YF vaccinations and on the JE/YF baseline serostatus of the subjects in this study. This information could also have potentially improved the predictive performance of the statistical models developed in this study.”*

Lines 281 & 282. The statement regarding mass vaccination campaigns needs to be corrected to indicate that The Philippines have stopped mass vaccination due to safety concerns.

Thank you for picking this up, we have now corrected the information as suggested.

Summary of changes:

- In the Discussion, we have corrected our statement as follows:
Lines 374-378: *“These results are reassuring in the context of the past immunization campaign conducted in the Philippines⁹ in 2017, which were stopped following the press release announcing the results obtained from new analysis of long-term data²⁰.”*

References

1. Efficacy and Safety of Dengue Vaccine in Healthy Children. ClinicalTrials.gov. [Accessed 19 May 2018] <https://clinicaltrials.gov/ct2/show/NCT00842530>
2. Study of a Novel Tetravalent Dengue Vaccine in Healthy Children Aged 2 to 14 Years in Asia. ClinicalTrials.gov. [Accessed 19 May 2018] <https://clinicaltrials.gov/ct2/show/NCT01373281>
3. Study of a Novel Tetravalent Dengue Vaccine in Healthy Children and Adolescents Aged 9 to 16 Years in Latin America. ClinicalTrials.gov. [Accessed 19 May 2018] <https://clinicaltrials.gov/ct2/show/NCT01374516>
4. Bhatt S, Gething PW, Brady OJ et al. The global distribution and burden of dengue. *Nature* 2013; **496**: 504–7.
5. Gilbert B, Golding N, Zhou H et al. Predicting the risk of avian influenza A H7N9 infection in live-poultry markets across Asia. *Nat Commun.* 2014; **5**: 4116.
6. Kraemer MUG, Sinka ME, Duda KA et al. The global distribution of the arbovirus vectors *Aedes aegypti* and *Ae. albopictus*. *eLife* 2015; 4: e08347
7. Sinka ME, Bangs MJ, Manguin S et al. A global map of dominant malaria vectors. *Parasit Vectors* 2012; 5:69.
8. Sinka ME, Bangs MJ, Manguin S et al. The dominant *Anopheles* vectors of human malaria in the Asia-Pacific region: occurrence data, distribution maps and bionomic précis. *Parasit Vectors* 2011; 4:89.
9. Pigott DM, Golding N, Mylne A et al. Mapping the zoonotic niche of Ebola virus disease in Africa. *eLife* 2014. 3: e04395.
10. Elith J, Leathwick JR, Hastie T, A working guide to boosted regression trees. *J Anim Ecol.* 2008; **77**(4): 802-13.
11. Elith J, Graham CH, Anderson RP et al. Novel methods improve prediction of species' distributions from occurrence data. *Ecography* 2006; 29: 129-151.
12. Sollich P. Probabilistic methods for Support Vector Machines. *Adv Neural Inf Process Syst* 2000; 1:349-355.
13. Tao Q, Wu G, Wang F, Wang J. Posterior probability support vector machines for unbalanced data. *IEEE Trans. Neural Netw.* 2005; 16(6): 1561-1573.
14. Freeman EA and Moisen G (2008). PresenceAbsence: An R Package for Presence-Absence Model Analysis. *Journal of Statistical Software*, 23(11):1-31.
15. Dorigatti I, Aguas R, Donnelly CA et al. Modelling the immunological response to a tetravalent dengue vaccine from multiple phase-2 trials in Latin America and South East Asia. *Vaccine* 2015; **33**(31): 3746-51.
16. Ferguson NM, Rodríguez-Barraquer I, Dorigatti I, Mier-y-Teran-Romero L, Laydon DJ, Cummings DAT. Benefits and risks of the Sanofi-Pasteur dengue vaccine: modelling optimal deployment. *Science* 2016; **353**(6303): 1033-1036.
17. Hadinegoro SR, Arredondo-García JL, Capeding MR et al. Efficacy and long-term safety of a dengue vaccine in regions of endemic disease. *N Engl J Med.* 2015; **373**(13): 1195-206.

REVIEWERS' COMMENTS:

Reviewer #2 (Remarks to the Author):

The authors did a reasonable job in revising the paper. I don't need to read this paper again. The authors can do a minor revision to address the comments below.

In lines 351-353 of the main text, this sentence does not read right, possibly due to some typos: "The statistically significant difference observed in the proportions of cases of cases with known and unknown PD3 titres between the subsets with known or unknown baseline serostatus"

Below are some minor comments for the authors to consider for the supplementary material.

For random search for hyper-parameter values, depending on the hyper-parameter, the value may need to be found through uniform sampling on the logarithmic scale if the possible value varies over a large range. For example, for support vector machine, the authors wrote "the cost of constraints violations (uniformly sampled between 0.01 and 10,000)", but typically this should be done on the logarithmic scale. There are several other hyper-parameters falling into this category.

The authors wrote that "We did not utilise automatic imputation technique to fill in for missing data in the predictors for the machine learning algorithms unable to handle incomplete data and in these cases we restricted the model training and evaluation to the subjects with complete data on the predictors." This is OK, but could be suboptimal. Usually one also tries to fill in the missing values (e.g., using missforest) and see how the results look like for these machine learning algorithms.

Point-by-point response to referee's comments and editorial requests

Refining the characterization of the Sanofi Pasteur dengue vaccine's efficacy profile using machine learning

Dorigatti I, Donnelly CA, Laydon DJ, Small R, Jackson N, Coudeville L, Ferguson NM

Reviewer #2 (Remarks to the Author):

The authors did a reasonable job in revising the paper. I don't need to read this paper again. The authors can do a minor revision to address the comments below.

In lines 351-353 of the main text, this sentence does not read right, possibly due to some typos: "The statistically significant difference observed in the proportions of cases of cases with known and unknown PD3 titres between the subsets with known or unknown baseline serostatus"

We thank the reviewer for pointing out that "of cases" was misplaced and that the sentence was not clear. We have clarified the sentence as detailed below.

Summary of changes:

- In lines 402 – 407, we revised the sentence as follows: "*The statistically significant difference observed in the proportions of cases with known or unknown baseline serostatus, among both subsets of subjects with known and unknown PD3 titres (p -values < 0.001, see Table 2), was due to the fact that the trial design specified that PD3 blood samples were retrospectively tested for dengue antibody levels for all dengue cases plus all participants in the immunological subsets of the trials.*"

Below are some minor comments for the authors to consider for the supplementary material.

For random search for hyper-parameter values, depending on the hyper-parameter, the value may need to be found through uniform sampling on the logarithmic scale if the possible value varies over a large range. For example, for support vector machine, the authors wrote "the cost of constraints violations (uniformly sampled between 0.01 and 10,000)", but typically this should be done on the logarithmic scale. There are several other hyper-parameters falling into this category.

We thank the reviewer for their insights and confirm that in our analysis we have uniformly sampled the parameters on the logarithmic scale when the range was large. For instance, the cost constraints mentioned above was sampled using: $\text{cost} = 10^{\text{sample}(c(-2,-1,0,1,2,4),1))}$ and we used the same rationale for the gamma and coef0 parameters. We have now revised the SI text and clarified the instances in which the logarithmic scale was applied.

Summary of changes:

- On page 50 of the SI, we revised the text as follows: "*for SVM, we also randomly sampled the kernel ('linear', 'sigmoid', 'polynomial' or 'radial'), the degree of the kernel for polynomial kernels (integer distribution between 1 and 10), the gamma kernel parameter (uniformly sampled between 0.01 and 100 on the logarithmic scale), the coef0 parameter used for polynomial and sigmoidal kernels (uniformly sampled between 1 and 1,000 on the logarithmic scale), the cost of constraints violations (uniformly sampled between 0.01 and 10,000 on the logarithmic scale) and class weights (uniformly sampled between 0.5 and 1 for baseline seronegative subjects, implying a higher weight than for baseline seropositive subjects); "*

The authors wrote that “We did not utilise automatic imputation technique to fill in for missing data in the predictors for the machine learning algorithms unable to handle incomplete data and in these cases we restricted the model training and evaluation to the subjects with complete data on the predictors.” This is OK, but could be suboptimal. Usually one also tries to fill in the missing values (e.g., using missforest) and see how the results look like for these machine learning algorithms.

We thank the reviewer for their opinion on the use of automatic imputation. As discussed in our previous response, one of the appealing properties of BRT is its ability to incorporate missing data in the predictors, implying that BRT does not require automatic imputation to fill in for missing data.

The reviewer’s suggestion applies to random forest, which was used in a sensitivity analysis. In our dataset, 18% (28 out of 155) of the cases and 2.5% (106 out of 4,096) of the non-cases with observed baseline serostatus had incomplete data on at least one of the other predictors (e.g. PD3 PRNT50 titres and the serotype of infection). The use of automatic imputation techniques to fill in for missing values (e.g. with the average observed across the dataset or with machine learning techniques, as would be the case with missforest) would be particularly challenging in this context due to the biological complexity of dengue infection (e.g. the large individual-level variability in PRNT titres) and the limited extent to which out-of-sample validation could be performed within each stratification, especially among cases. Imputing the baseline serostatus (response variable) using predictors that have been filled in with imputations would risk propagating biases and errors in unintuitive ways and while we strongly believe that data imputation is a powerful tool to gain useful insights into biological processes, we also advocate for the most transparent and validated use of machine learning for data imputation. This is particularly important given that the estimates produced in this analysis will affect the future use and deployment of the CYD-TDV vaccine.